



# Radar observations of winds, waves and tides in the mesosphere and lower thermosphere over South Georgia island (54°S, 36°W) and comparison to WACCM simulations

Neil P. Hindley[1], Neil Cobbett[2], Dave C. Fritts[3], Diego Janchez[4], Nicholas J. Mitchell[1,2], Tracy Moffat-Griffin[2], Anne K. Smith[5], and Corwin J. Wright[1]

[1]Centre for Space, Atmospheric and Oceanic Science, University of Bath, Bath, UK
[2]British Antarctic Survey, Cambridge, UK
[3]GATS, Boulder, CO, USA
[4]NASA Goddard Space Flight Center, Greenbelt, MD, USA
[5]National Center for Atmospheric Research, Boulder, CO, USA

**Correspondence:** n.hindley@bath.ac.uk

**Abstract.** The mesosphere and lower thermosphere (MLT) is a dynamic layer of the earth's atmosphere. This region marks the interface at which neutral atmosphere dynamics begin to influence the ionosphere and space weather. However, our understanding of this region and our ability to accurately simulate it in global circulation models (GCMs) is limited by a lack of observations, especially in remote locations. To this end, a meteor radar was deployed on the remote mountainous island
of South Georgia (54°S, 36°W) in the Southern Ocean from 2016 to 2020. The goal of this study is to use these new measurements to characterise the fundamental dynamics of the MLT above South Georgia including large-scale winds, solar tides, planetary waves (PWs) and mesoscale gravity waves (GWs). We first present an improved method for time-height localisation of radar wind measurements and characterise the large-scale MLT winds. We then explore the amplitudes and phases of the diurnal (24h), semidiurnal (12h) and terdiurnal (8h) solar tides at this latitude. We also explore PW activity and find very large
amplitudes up to 30 ms⁻¹ for the quasi-2 day wave in summer and show that the dominant modes of the quasi-5, 10 and 16 day waves are westward W1 and W2. We investigate wind variance due to GWs in the MLT and use a new method to show an east-west tendency of GW variance of up to 20% during summer and a weaker north-south tendency of 0-5% during winter. This is contrary to the expected tendency of GW directions in the winter stratosphere below, which is a strong suggestion of secondary GW (2GW) observations in the MLT. Lastly, comparison of radar winds to a climatological Whole Atmosphere Community
Climate Model (WACCM) simulation reveals a simulated summertime mesopause and zonal wind shear that occur at altitudes around 10 km lower than observed, and southward winds during winter above 90 km altitude in the model that are not seen in observations. Further, wintertime zonal winds above 85 km altitude are eastward in radar observations but in WACCM they are found to weaken and reverse to westward. Recent studies have linked this discrepancy to the impact of 2GWs on the residual circulation which are not included in WACCM. These measurements therefore provide vital constraints that can guide the
development of GCMs as they extend upwards into this important region of the atmosphere.



## 1 Introduction

The mesosphere and lower thermosphere (MLT) is an atmospheric region that marks the transition between the neutral dynamics of the middle atmosphere and ionised processes in the thermosphere and ionosphere above (Smith et al., 2011, 2017; Jackson et al., 2019; Sassi et al., 2019). The unique features of this region set the MLT apart from other atmospheric layers
(Smith, 2012), including the coldest naturally occurring temperatures at the summertime polar mesopause, enormous local dynamical variability due to atmospheric tides and planetary waves (PWs), and a residual circulation that is to first order driven by small-scale atmospheric gravity waves (GWs).

The dynamics and circulation in the MLT are important for global transport of important trace chemical species (Smith et al., 2011; Kvissel et al., 2012), including transporting $NO_x$ and meteor smoke into the winter polar stratosphere which can affect
stratospheric ozone and surface climate (Funke et al., 2017; Garcia et al., 2017), while the annual formation and depletion of polar mesospheric clouds (PMCs) at the summertime mesopause has impacts on sensitive chemical processes (Thurairajah et al., 2013; Siskind et al., 2018). Neutral winds in the MLT can also have first-order effects on the impacts of space weather in the ionised atmosphere above (Jackson et al., 2019; Sassi et al., 2019).

The MLT is also where the impact of solar tides on the neutral winds is greatest. Direct solar heating of the stratosphere
below causes tides to propagate upwards and grow exponentially in amplitude, leading to wind reversals in the MLT that can be up to $100 \, \text{ms}^{-1}$ over the course of one day (e.g. Jacobi et al., 1999; Mitchell et al., 2002; Murphy et al., 2006; Vincent, 2015). Planetary scale waves, which occur at periods from 2 days to more than 16 days and can reach amplitudes up to several 10s of $\text{ms}^{-1}$ (Schoeberl and Clark, 1980; Salby, 1981a, b), also play a key role in the dynamics of the MLT by modulating GW breaking (Holton, 1984) and through non-linear interactions with solar tides (Beard et al., 1999).

One fundamental aspect of the MLT is its strong response to the forcing due to atmospheric GWs, which results in upwelling in the middle atmosphere over the summertime pole and downwelling over the winter pole (Soloman and R., 1987; Vargas et al., 2015). These adiabatic cooling and heating conditions drive the thermal structure of the atmosphere away from that expected under radiative equilibrium, leading to a global-scale pole-to-pole residual circulation from the summer pole to the winter pole (Houghton, 1978; Holton, 1983; Becker, 2012).

The sensitivity of the residual MLT circulation to GWs makes its simulation in high-top global models especially challenging (Becker, 2012; Yasui et al., 2018; Jackson et al., 2019). The majority of GWs in global models and their generation mechanisms are sub-grid scale, and the momentum deposition and subsequent driving due to these waves must be parameterised (Alexander et al., 2010). In nearly all current GW parameterisations however, the magnitude and direction of GW momentum that reaches the MLT is almost entirely dependent on the selected GW launch spectrum near the surface, the vertical columnar propagation
of GW momentum and filtering by the background winds below. Circulations in the MLT can therefore be highly sensitive to the tuning of these GW parameterisations in ways that the lower atmosphere is not. Further, concepts like oblique GW propagation (e.g. Kalisch et al., 2014) and/or secondary GWs (Vadas et al., 2018; Vadas and Becker, 2018) can change the magnitude and direction of GW momentum reaching the MLT and are not currently included in the standard parameterisations in operational global models.



Developments of advanced models employing realistic parameterisations of subgrid-scale GW influences and high time cadence observations of neutral winds, waves and tides in the MLT are required to make progress in this regard. Satellite observations can provide a global picture, but they lack the sampling cadence to accurately constrain short timescale variability of GW processes. Meteor wind radars (MWRs) however offer one of the best methods for measuring the neutral winds in the MLT. By measuring the radial Doppler shift of reflected radio pulses from ionised meteor trails near 90 km altitude, MWRs

can derive continuous measurements of the neutral winds in the MLT at one location (Hocking et al., 2001).

To this end, a meteor radar was installed on remote island of South Georgia (54°S, 36°W) in the Southern Ocean. The radar made near-continuous measurements of neutral winds in the MLT from February 2016 to November 2020. South Georgia is located near to the global GW "hot spot" of activity in the stratosphere of the southern Andes and Antarctic Peninsula (Hindley et al., 2015; Hoffmann et al., 2013; Hindley et al., 2020), and is also an intense source of wintertime GW activity itself

(Hoffmann et al., 2014; Hindley et al., 2021). Further, recent observations and modelling have indicated significant generation and propagation of 2GWs in the mesosphere and thermosphere in the region (Becker and Vadas, 2018; Kogure et al., 2020; Lund et al., 2020; Fritts et al., 2021).

The goal of this study is to use these measurements to characterise the fundamental dynamics of the MLT at this remote location. In Sect. 2 we describe the radar, satellite and modelling data sets used in this study, and in Sect. 3 we describe a

new method for localisation of derived radar winds. There then follow five results sections: in Sect. 4 we show mean winds and temperatures over the island; in Sect. 5 we characterise the solar tides; in Sect. 6 we investigate PWs; and in Sect. 7 we investigate GW activity. Lastly, in Sect. 8 we compare observed winds and temperatures in the MLT to climatological dynamics from WACCM. Our results are discussed in Sect. 9 and we summarise the study and draw our conclusions in Sect. 10.

## 2   Data

### 2.1   The South Georgia meteor radar


A SKiYMET VHF meteor radar was installed at King Edward Point (KEP) on the island of South Georgia (54°S, 36°W) in the Southern Ocean in January 2016. The radar was deployed in an "all sky" configuration and consists of a single solid-state transmitter operating at 35.24 MHz with a pulse repetition frequency (PRF) of 625 Hz and 7-bit Barker code, and a five element receiver array. It was equipped with an interferometer for simultaneous measurement of range, zenith angle and azimuth of

ionised meteor trials that enables the height and location of these trails to be determined. The Doppler shift of the returning radio pulses can be used to infer a radial "drift velocity" for each detected trail, which can be interpreted as a radial wind vector measurement at the given height, location and time. For a full description of the SKiYMET meteor radar system, see Hocking et al. (2001).

An overview of the meteor detections by the KEP radar is provided in Fig. 1. Radial drift velocities are measured between

heights of around 70 to 110 km altitude (Fig. 1b), centred near 90 km. The peak height of the meteor distribution is related to neutral density and can vary seasonally by a few km.

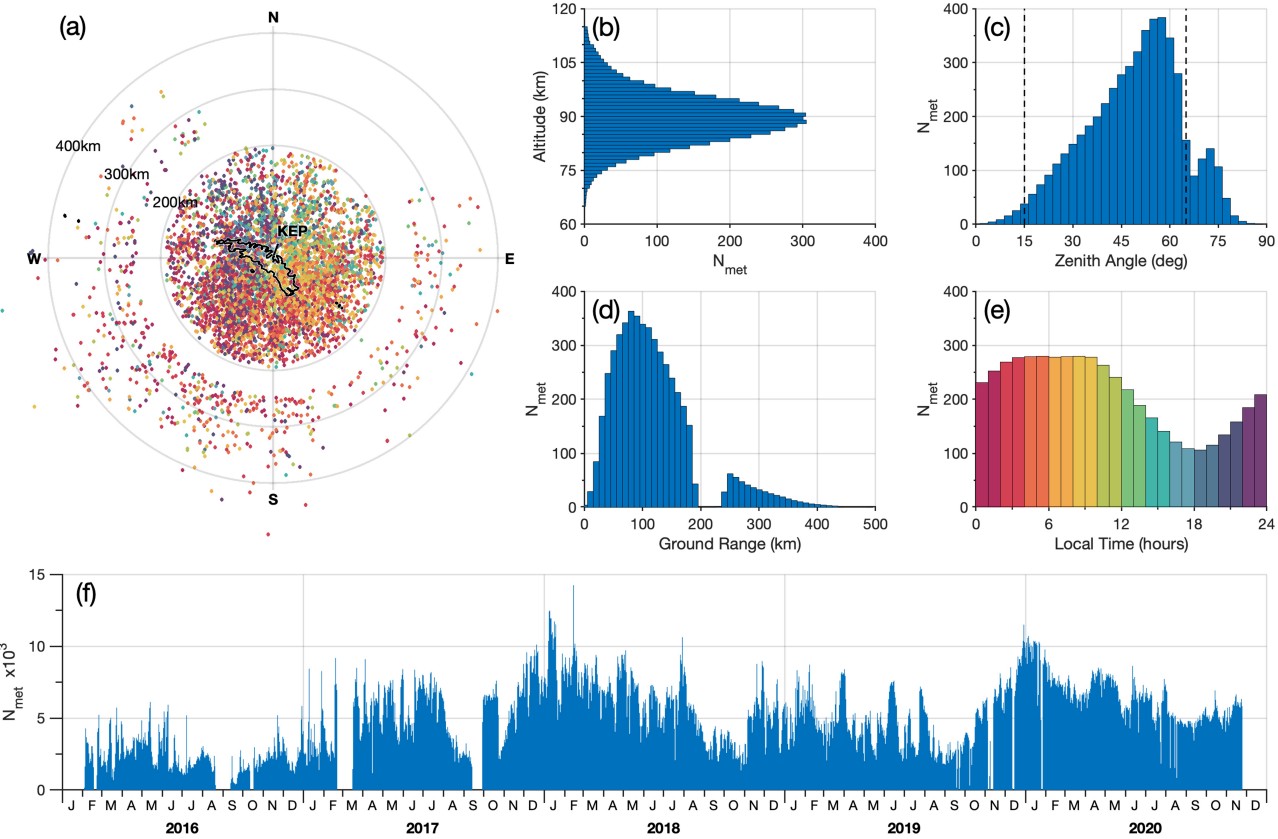

**Figure 1.** Distributions of meteor echoes detected over King Edward Point (KEP) on South Georgia. Panel (a) shows the horizontal distribution of meteor echoes on 21st June 2018, where echoes are coloured according to their time of detection (see panel e for colour scale), while panels (b), (c), (d) and (e) show histograms of the average height, zenith angle, horizontal (ground) range and local time of all detected meteor echoes respectively per day for all operational days. Panel (f) shows the number of meteor echoes detected per day for each day of operation during 2016 to 2020.

Meteors are detected at angles up to ∼80° from the zenith (Fig. 1c), however in this study we only select meteors between zenith angles of 15 and 65 degrees (dashed lines in Fig. 1c). This is to avoid potential errors in the projection of horizontal wind from meteor echoes near the zenith, and errors in the measured height of meteor echoes at large zenith angles.

Meteor are detected in all horizontal directions as illustrated in Fig. 1a. The receiver antennae are switched off during each transmitter pulse to avoid over saturating the receiver array, resulting in a small band of arrival times during which reflected radio pulses are not detected. This results in the horizontal ring between 200 to 250 km from the radar where no drift velocity measurements can be made. Despite the island's highest mountains lying to the west and south, there is also an obstruction of detections at large zenith angles to the north east due to the proximity of the relatively small Mount Duce.



The number of unambiguous meteor detections per day is shown in Fig. 1f. The radar began collecting data on 3rd February 2016, typically detecting between around 2000 to 5000 meteor echoes per day. This increased to around 4000 to 8000 meteors per day from 2017 onwards. There are some time periods during 2016, 2017 and 2019 when the radar had be taken offline due to power limitations at KEP base, which is supplied by a hydroelectric plant on the island, however the use of the hydroelectric energy prevented any unwanted interference from power generators on the base. The radar was uninstalled on 25th November

2020 and is expected to be redeployed at Halley research station in Antarctica.

In this study we also use measurements from the Southern Argentina Agile Meteor Radar (SAAMER Fritts et al., 2010b, a) system deployed at Rio Grande (54°S, 68°W) in Tierra del Fuego, Argentina. The SAAMER radar is located around 2000 km to the west of South Georgia at the same latitude, which provides an opportunity for an investigation into eastward and westward propagating planetary wave modes in Sect. 6. Throughout the study, identical data processing steps to derive winds are applied

to the meteor detections from the SAAMER and KEP radars for consistency. Derived winds from the KEP radar have also contributed to the studies of Liu et al. (2021) and Stober et al. (2021a).

## 2.2   MLS

Here we use version 5 of the level 2 temperature retrieval from the Microwave Limb Sounder (MLS, Waters et al., 2006) that flies aboard NASA's Aura satellite. Aura was launched in 2004 and is part of the "A-train" constellation, following a sun-

synchronous polar orbit and crossing the equator at 0130 and 1330 local time each day (Schoeberl et al., 2006). MLS measures vertical profiles of microwave emissions of the atmospheric limb in five spectral bands over the altitude range of approximately 261 to 0.001 hPa (around 10 km to 100 km). Temperature and pressure are retrieved from the 118 and 239 GHz bands with an estimated temperature precision in the middle atmosphere better than 3-4 K and an accuracy of between 2-3 K (Livesey et al., 2015; Schwartz et al., 2008). The vertical resolution of MLS varies from around 3.6-5 km between 10 to 25 km altitude to

greater than 5 km above 40 km altitude, and the along-track spacing of the vertical profiles is approximately 170 km.

## 2.3   WACCM

The Whole Atmosphere Community Climate Model (WACCM) is a comprehensive global climate model that extends from near the surface to the lower thermosphere, at around 140 km altitude. Here we use an ensemble of three WACCM simulations for the period 1950-2014 that have specified sea surface temperatures based on observations. Other external input such as

anthropogenic pollutants and volcanic emissions are also based on the observational records. The atmospheric simulations are free-running and coupled to interactive chemistry and radiation. This specific configuration is part of the Coupled Earth System Model (CESM) version 2 (Danabasoglu et al., 2020) and was completed as a contribution to the sixth round of the Coupled Model Intercomparison Project (CMIP6, Eyring et al., 2016) using the latest version (version 6) of the model (Gettelman et al., 2019). Improvements of WACCM6 over previous versions include a finer horizontal grid (0.95 × 1.25 degrees), improved

atmospheric chemical processes and aerosols (Tilmes et al., 2019), an expanded database of volcanic eruptions (Neely III, R.R and Schmidt, 2016), additional fluxes of energetic particles due to space weather (Marsh et al., 2007; Matthes et al., 2017) and





realistic magnitudes and occurrences rates of the Quasi-Biennial Oscillation (QBO) and El Niño Southern Oscillation (ENSO). For a detailed description of WACCM version 6 and its validation, we refer to Gettelman et al. (2019).

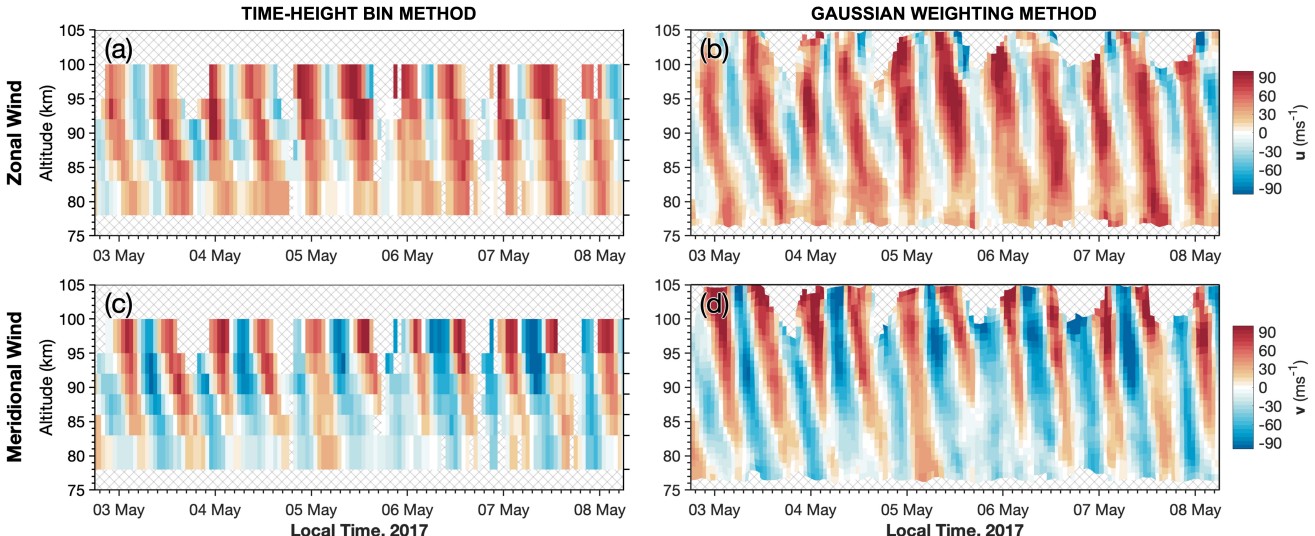

**Figure 2.** Derived zonal and meridional winds from the South Georgia meteor radar during May 2017 using (a,b) a traditional time-height binning approach and (c,d) a Gaussian time-height weighting approach described here. Hatched areas show regions with too few meteors to reliably derive winds.

## 3 Methods

### 3.1 Improved time-height localisation of radar winds using a Gaussian weighting approach

Time-height localised measurements of zonal and meridional winds $u$ and $v$ from meteor radar systems are usually derived by binning the measured radial velocities of individual meteors into time-height bins (e.g. Hocking et al., 2001). Then, for all meteor echoes $i = 1, 2, \ldots, N$ in each bin, a least-squares fit of the function

$$
\begin{pmatrix} v_{h_1} \\ v_{h_2} \\ \vdots \\ v_{h_N} \end{pmatrix} = \begin{pmatrix} \sin \theta_1 & \cos \theta_1 \\ \sin \theta_2 & \cos \theta_2 \\ \vdots & \vdots \\ \sin \theta_N & \cos \theta_N \end{pmatrix} \begin{pmatrix} u \\ v \end{pmatrix} + \begin{pmatrix} \varepsilon_1 \\ \varepsilon_2 \\ \vdots \\ \varepsilon_N \end{pmatrix}.
\tag{1}
$$

is performed to recover estimates of $u$ and $v$, where $v_h = v_r / \sin \phi$ is the horizontal projection of the measured radial velocity $v_r$ of the meteor trail, $\phi$ is the angle from the zenith, and $\theta$ is azimuth (defined clockwise from north), and $\varepsilon$ is an error term





for each meteor measurement. These time-height bins can be typically around 1-2 hours in time and $\sim$3 km in height (Hocking and Thayaparan, 1997; Mitchell et al., 2002; Mitchell and Beldon, 2009). A threshold value of at least 20 meteor echoes in a bin can be applied to ensure a reliable fit (Mitchell and Beldon, 2009).

This method is simple and effective, but it has several important limitations, namely (1) meteors near the boundaries of the time-height bin count 100% to the centre of the bin, but not to neighbouring bins; (2) the fit is not constrained by the fits of neighbouring bins, that is, we would not expect a wildly different value for $u$ and $v$ from one hour to the next to be physical, yet this is permitted by the method; and (3) in bins with low meteor counts the cutoff threshold is applied and the fit is not performed, even if the fit could be constrained by using neighbouring bins.

Here we describe a new approach to mitigate these problems. Instead of defining a time-height bin centred at time $t_0$ and height $z_0$, we define a 2-dimensional Gaussian weighting function centred at $(t_0, z_0)$. This Gaussian function has standard deviations $\sigma_t = 0.85$ h and $\sigma_z = 1.275$ km, which correspond to full-width-at-half-maximum (FWHM) values of 2 hours and 3 km in time and height respectively. The central location of the function $(t_0, z_0)$ is then moved through time and height in steps of 1 hour and 1 km. For a given height and time at each of these steps, each meteor echo $i$ has a weighting $w_i$, which is 150    the product of its weightings in time $w_{ti}$ and height $w_{zi}$, given by

$$
\begin{aligned}
w_i &= w_{ti} \times w_{zi} \\
&= \exp\left(\frac{-(t_i - t_0)^2}{2\sigma_t^2}\right) \times \exp\left(\frac{-(z_i - z_0)^2}{2\sigma_z^2}\right)
\end{aligned}
\tag{2}
$$

These weightings are then used to perform a weighted least-squares fit of the function in Eqn. 1. The fit is performed using a weighted matrix inversion method as

$\begin{pmatrix} u \\ v \end{pmatrix} = \left(X^\top \cdot (W_2 X)\right)^{-1} \cdot X^\top \cdot (W_1 Y)$    (3)

where $X$ is an $N \times 2$ matrix containing the sine and cosine terms of azimuth (as in Eqn. 1), $W_1$ is an $N \times 1$ vector containing the combined time-height weightings $w_i$, $Y$ is an $N \times 1$ vector containing the measured horizontal velocities $v_{h_i}$. The dot $\cdot$ denotes matrix multiplication and $X^\top$ denote the transpose. $W_2$ is simply an $N \times 2$ duplicate of the weighting vector $W_1$. Note that this weighted inversion can be arranged analytically in several different ways, but this approach was found to have the 160    smallest computational expense because it minimises the dimensions of the matrices involved and avoids an $N \times N$ weighting matrix for $W$.

To further improve computation speed, we only consider horizontal velocities from meteor echoes with a combined weighting $w_i > 0.05$ (around two standard deviations) for each fit, which keeps the above matrices relatively small. This cut off means that winds derived at least two standard deviations apart in height ($\pm 1.7$ km) or time ($\pm 2.55$ hours) are entirely independent.

Due to the irregular distribution of meteors in time and height, for each fit we use the same weights in the vector $W_1$ to compute a weighted mean of the altitude and time to which the derived winds correspond. This means that our derived horizontal winds vary smoothly and accurately with the true distribution of meteor detections in height and time, which is not necessarily the case with the traditional time-height bin approach where fixed bin centres are used (Mitchell et al., 2002).





## 3.2 Comparison to a traditional height gates approach

The new Gaussian weighting method is compared to a traditional time-height bin method in Fig. 2. Derived zonal (top) and meridional (bottom) winds from radar measurements at KEP during the period 3rd to 8th May 2017 are shown for a time-height bin method (left) and the new Gaussian weighted method describe above (right).

The height bins chosen follow those used by Mitchell et al. (2002) and are between altitudes of 78-83 km, 83-86 km, 86-89 km, 89-92 km, 92-95 km and 95-100 km. They are 2 hours wide in time and are stepped along in 1 hour steps. A threshold
of at least 20 meteors for the fit is applied in both methods, and the hatched regions indicate where this condition is not met.

Periodic oscillations near 12 hours in the zonal and meridional winds are found in Fig. 2 in both methods. This is the semidiurnal solar tide, which is dominant at this latitude and season.

Several advantages to the new Gaussian weighting method are found. Firstly, the full altitude range of the available measurements is revealed and reliable winds are automatically found at higher and lower altitudes during the morning (local time)
when meteor counts are high. On some occasions, we found that winds can be derived up to 110 km altitude (not shown) with a realistic tidal phase progression with height, suggesting that altitude independence can be maintained with this method despite low meteor counts.

Secondly, winds are successfully derived by the Gaussian weighting method for time periods where too few meteors were detected to ensure a reliable fit using the traditional method, such as during the afternoon of 6th May 2017. This is because
additional meteor echoes, which would have been in adjacent bins in the traditional method, are available to constrain the fit in Eqn. 3. These missing periods cannot be filled in simply by the interpolation of the surrounding wind measurements. This also has the advantage of reducing spurious wind fits during periods of low meteor counts, because the inclusion of neighbouring meteor measurements helps to prevent unphysically large changes in wind between two adjacent time steps or height levels.

Finally, we note that the wind measurements in the Gaussian weighting method appear to warp away from a regular time-
height grid at upper and lower altitudes. This is due to the derived winds being correctly allocated to the weighted mean time and height of the available meteors, which is not considered in the traditional method.

This new approach can therefore be useful for considering the full vertical extent of meteor radar winds at short time scales, and is applicable to any meteor wind radar system. As mentioned above, if the radar system is powerful enough to detect sufficient numbers of meteors, these winds could be reliably derived at even finer time-height scales.

## 4 Large-scale winds and temperatures in the MLT over South Georgia

The general dynamics of the large-scale zonal and meridional winds over South Georgia are characterised in the power spectra in Fig. 3, which shows a normalised Lomb-Scargle periodogram of hourly winds at 90 km altitude for the period February 2016 to November 2020 inclusive. The large semidiurnal solar tide $S_2$ dominates, with the diurnal ($S_1$) and terdiurnal ($S_3$) tides showing roughly equal amplitudes at this latitude. At periods longer than 1 day, peaks near 2, 5, 10 and 16 days are found
which correspond to known planetary wave periods, although other periodicities are also present (e.g. quasi-6 days). At periods shorter than the inertial period ($f \sim 14.8$ hours at this latitude) but excluding tidal periods, the spectra are dominated by gravity





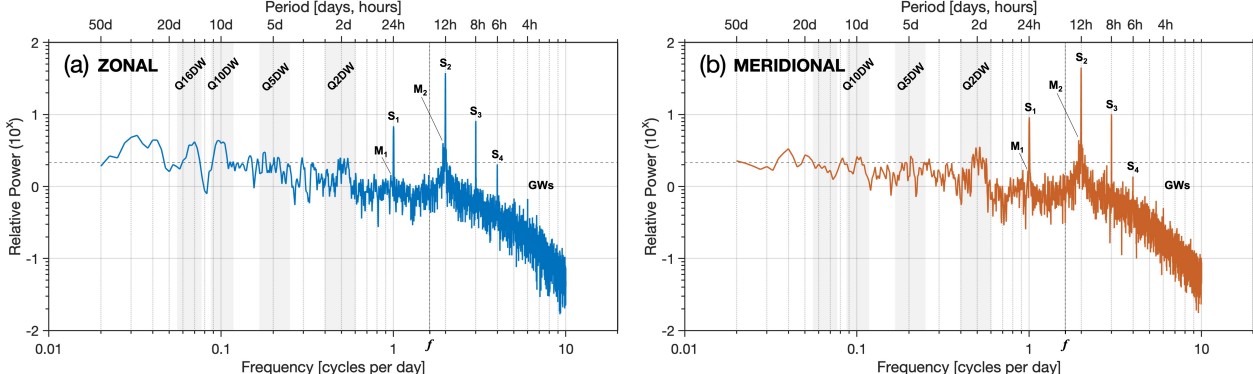

**Figure 3.** Normalised power spectra of the hourly zonal (a) and meridional (b) winds at 90 km altitude measured by the South Georgia meteor radar for the period 2016 to 2020. Annotations show peaks corresponding to solar tides $S_{1-4}$, lunar tides $M_{1-2}$ and planetary waves, where shaded grey regions for the latter show an approximate range of periods. The dashed horizontal line shows the 90% confidence level, and $f$ denotes the inertial period (∼14.8 hours) at 54°S.

waves and closely follow a $-5/3$ gradient as expected from theory (e.g. Smith et al., 1987). A significant peak is also found near 12.4 hours which likely corresponds to the semidiurnal lunar tide $M_2$, suggesting that this South Georgia radar dataset could be useful for investigating lunar tides, although the diurnal lunar tide $M_1$ appears to be very weak at this latitude.

To explore the dynamics of the MLT on monthly timescales, Fig. 4 shows derived monthly zonal (top) and meridional (bottom) winds against altitude for 2016 to 2020. These winds are derived as monthly composite days where, for each month, all meteor measurements in that month are assumed to occur on the same day. Hourly winds are fitted via the method in Sect. 3.1 and the average wind for each monthly composite day is found. The rightmost panels in Fig. 4 show a composite year using the same 30-day sliding window, except that all measurements are assumed to occur in the same year.

The zonal winds in Fig. 4(a,b) indicate a clear annual cycle, with eastward winds in austral winter and a strong wind shear that descends in altitude during summer. The onset and descent of this summertime wind shear follows a characteristic "triangular" pattern when plotted on time-height axes, with a rapid (∼1 month) onset during spring followed by a gradual descent in altitude throughout summer into autumn over approximately 5 to 6 months. This is consistent with other meteor radar wind observations at high latitudes in both hemispheres (e.g. Fritts et al., 2010b; Sandford et al., 2010; Stober et al.,

2021b). The vertical gradient of the zonal wind in summer is particularly strong, from around -20 ms$^{-1}$ (westward) to up to +40 ms$^{-1}$ (eastward) over only 10 to 15 km altitude.

In the monthly mean winds shown here, there does not appear to be a significant change in the observed zonal winds over South Georgia during the southern sudden stratospheric warming (SSW, e.g. Rao et al., 2020) in September 2019. This is likely because, as shown recently by Liu et al. (2021), wind responses occurred on shorter timescales of up to around 10 days,

and were also co-incident with a large amplitudes of the quasi-6 day planetary wave.

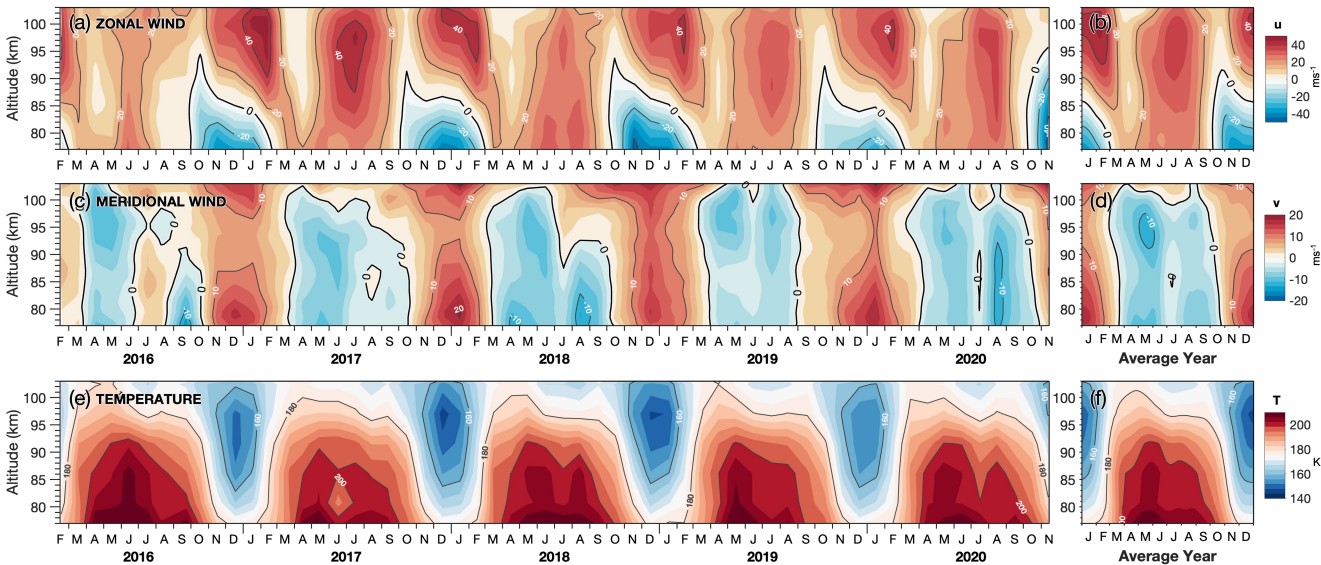

**Figure 4.** Monthly mean zonal and meridional winds (a-d) in the MLT from radar measurements over South Georgia during 2016 to 2020. Panels (e,f) show monthly mean temperatures from MLS satellite measurements. Rightmost panels show winds and temperatures derived from an average (composite) year using all meteor detections and all temperature measurements during 2016 to 2020.

Meridional winds in Fig. 4(c,d) are northward and up to $20\,ms^{-1}$ during summer, and southward wind during winter. This flow is part of the mesospheric branch of the Brewer-Dobson circulation (e.g. Butchart, 2014), as described by Murgatroyd and Singleton (1961). The summertime northward flow maximises near $80\,km$ altitude, while the wintertime southward flow is weaker but persists for approximately 1 to 2 months longer than the summertime conditions. Interestingly, during 2016, 2017 and 2020 there is a brief reversal of the southward flow to northward during winter. This also occurs at altitudes above $90\,km$ and $95\,km$ during 2018 and 2019 respectively. This weak wintertime reversal was also found in radar observations at high latitudes in both hemispheres by Sandford et al. (2010), which could suggest a weak semiannual modulation of the mesospheric branch of the Brewer-Dobson circulation. This modulation could be connected to the mesospheric component of the tropical semiannual oscillation (SAO, e.g. Smith et al., 2017; Ern et al., 2021). The weak meridional wind reversal seen here during winter coincides with the eastward phase of the tropical SAO in the mesosphere, which is known to be out of phase with the stronger stratospheric component of the SAO. Another explanation for this could be due to shifts in the amplitude and/or phase of quasi-stationary planetary waves in the winter stratosphere and lower mesosphere, which could play a major role in modulating the flow observed at a single location.

Observed winds during the winter of 2016 show differences to the other years in this five year period. The zonal winds are more than $10\,ms^{-1}$ weaker on average below $90\,km$ altitude during July-August, and more than $20\,ms^{-1}$ weaker than the following year during June-July above $90\,km$ altitude. Meridional winds during 2016 also show the largest wintertime reversal from southward to northward and back to southward again during June to October. Although meteor counts were lower than





other years (Fig. 1f), the sliding 30-day composite window still provides at least ∼30000 measurements from which to fit winds in a given composite day, so our results are unlikely to be affected by this.

The reason for this not immediately clear. It is known that the El Niño Southern Oscillation (ENSO) and Quasi-Biennial Oscillation (QBO) can have a significant effect on MLT dynamics (e.g. de Wit et al., 2016; Laskar et al., 2016; Sun et al., 2018). In 2016, ENSO was in an unusually strong warm phase, and there was an unprecedented disruption of the QBO (Osprey et al., 2016). As mentioned above, it is also likely that changes in quasi-stationary planetary waves in the winter hemisphere play a major role in the observed interannual variability. A study that could explore a possible link between these dynamical

processes and the reduced wind speeds observed over South Georgia in 2016 could provide valuable information into coupling processes between atmospheric layers. Such a study would however require a longer timeseries of observations for sufficient statistics and a numerical modelling component for explore a plausible mechanism, so is out of scope here.

The bottom panels of Fig. 4 show monthly mean temperatures in the MLT from MLS satellite observations, averaged over a horizontal region within a 400 km radius of the island for each height level, which is close to the size of the horizontal

collecting area of the meteor radar. These show the annual cycle of temperatures in the MLT, including the cold summertime mesospause where temperatures fall below 160 K between 95 and 100 km altitude around the summer solstice, despite the polar mesosphere being subject to constant sunlight. This temperature structure is different from that expected under radiative equilibrium (Geller, 1983), and is instead driven away from this state by the effects of atmospheric waves (e.g. Smith, 2012; Becker, 2012). The wintertime mesopause is warmer and occurs above 105 km altitude each year.

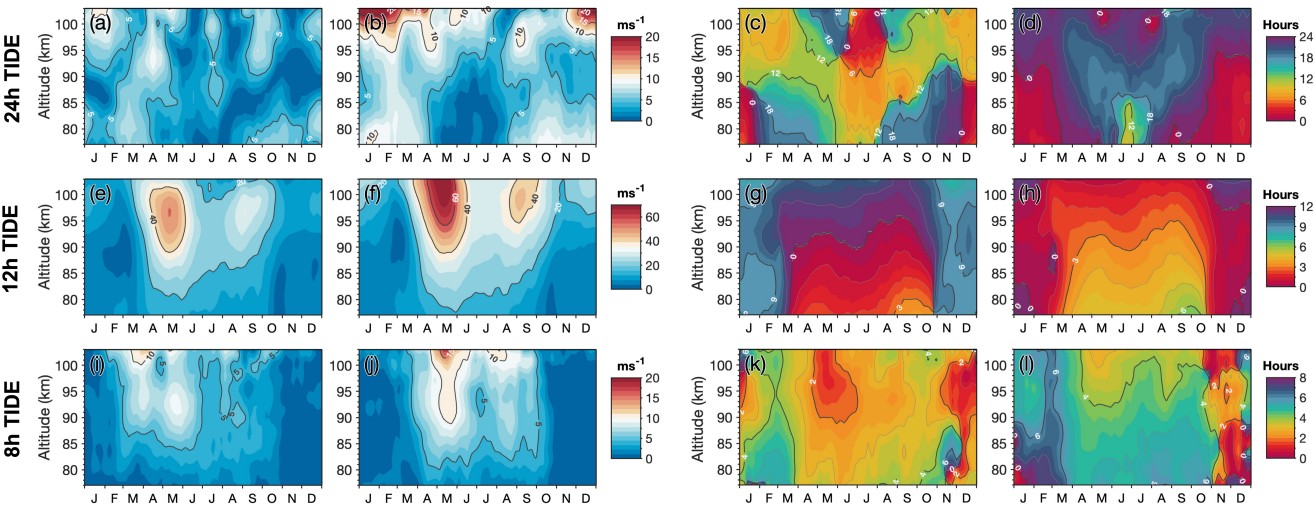

**Figure 5.** Zonal and meridional wind amplitudes (left) and phases (right) of the diurnal (24h), semidiurnal (12h) and terdiurnal (8h) solar tides in the MLT over South Georgia. Tidal periods are fitted to sliding 30-day composite days for all meteor measurements during 2016 to 2020 to make the average composite year shown. Phases are given in hours since midnight local time.





## 5    Solar Tides

The diurnal variability of the MLT region is dominated by solar tides, which have periods that are integer fractions of 1 day. Tides measured in the MLT have typically propagated upwards from the stratosphere and troposphere below, where the atmospheric response to solar heating is largest, rather than being directly excited in the MLT (Smith, 2003, 2012). As shown recently by Dempsey et al. (2021), tidal amplitudes and variability at high latitudes are not always well simulated in numerical models, so characterising the tidal magnitudes and seasonal variability here can have value.

To characterise the solar tides, sinusoids with periods of 24, 12, 8 and 6 hours are fitted to each height level of daily zonal and meridional winds. Although we apply a fit for the quardiurnal (6h) tide, we found that its amplitude too small to provide a satisfactory signal-to-noise in our analysis (see Fig. 3), but we acknowledge that the 6h tide can sometimes reach significant amplitudes at high latitudes (Smith, 2004).

Figure 5 shows the amplitudes and phases of the zonal and meridional components of the diurnal, semidiurnal and terdirunal solar tides. The results shown in Fig. 5 are for an average (composite) year, where the tidal periods are fitted to a sliding 30-day composite day of winds using measurements from the corresponding days in all years, as in Figs. 4(b,d). We found that this is more robust than fitting individual daily winds and taking an average, due to measurement gaps and spurious wind derivations that may occur during periods with few meteors. Monthly composite amplitudes and phases for the period 2016 to 2020 are included in supplementary Figs. S1 and S2.

The dominant tide in the MLT at this latitude is the semidiurnal tide (e.g. Murphy et al., 2006; Conte et al., 2017), as shown in Fig. 5, with amplitudes that can reach $60\,\text{ms}^{-1}$ and $80\,\text{ms}^{-1}$ in the zonal and meridional directions respectively during April to May each year. The intra-annual variability of the semidiurnal tide is broadly characterised by an "equinox high" and "solstice low", although we find a lag of around 1.5 months in the tidal maxima after the solstice and equinox dates. Consistently, we find that the meridional component of the semidiurnal tide is consistently around 25% to 35% larger than the zonal component here.

The amplitudes of the diurnal and terdiurnal tides in Figs. 5a,b and 5i,j are weaker, reaching values up to 15 and $10\,\text{ms}^{-1}$ respectively. The diurnal tide maximises in summer, where the meridional component is as much as three times as large as the zonal. Largest average values for the meridional component are up to $15\,\text{ms}^{-1}$ above $100\,\text{km}$ altitude during December to March, but can be as large as $25\,\text{ms}^{-1}$ during individual years. The altitude structure of the diurnal tide is also somewhat opposite to the semidiurnal, maximising above $100\,\text{km}$ and below $85\,\text{km}$, whereas the semidiurnal tide maximises in between these altitude ranges.

The amplitude of the terdiurnal tide broadly follows the annual and semiannual cycle of the semidiurnal tide, especially in the meridional direction in Fig. 5j, with maxima of around 5 to $10\,\text{ms}^{-1}$ around May and September. The autumnal maximum of the terdiurnal tide is broadly consistent with averaged amplitudes from December to March from satellite measurements by Smith (2000) at latitudes near 60°S. The timing of this maximum is an interesting result, because it has been suggested (Moudden and Forbes, 2013) that the observed terdiurnal tide could primarily be the result of nonlinear interaction of the diurnal (D) and semidiurnal (S) tides, rather than by direct solar excitation. Moudden and Forbes (2013) used global satellite measurements





from SABER to decompose the terdiurnal tide (T) into its constituent eastward (E) and westward (W) propagating modes and zonal wavenumbers (1, 2, ... N). They found that the dominant mode of the terdiurnal tide was TW3 which could be the result a non-linear interaction between DW1 and SW2.

However, as discussed in detail by Lilienthal et al. (2018), excitation mechanisms of the terdiurnal tide, in particular the role of direct solar heating versus non-linear interactions is still under debate. In the modelling studies of Lilienthal et al. (2018) and Lilienthal and Jacobi (2019), they found that the solar heating was dominant, but that non-linear interactions could become significant at midlatitudes during winter. The wintertime maximum of the terdiurnal tide in our results could therefore be consistent with the results of both Moudden and Forbes (2013) and Lilienthal and Jacobi (2019) and suggest that both factors could be important during this time.

Phases of the solar tides are shown on the right hand axes of Fig. 5, defined as the local time of the first eastward wind maximum in hours. When the amplitudes of each tide are large, the measured phases exhibit $\pi/2$ phase shift between the zonal and meridional components, consistent with circular polarisation of the tidal wind vector. Further, the meridional phase lags behind the zonal phase in time by one quarter cycle, indicating an anticlockwise rotation of the wind vector when considered in a hodograph context, which is indicative of an upwardly propagating tide in the southern hemisphere. However, when tidal amplitudes are weak, such as during winter for the diurnal tide and during summer for the terdiurnal tide, this phase relationship breaks down, indicating that our tidal analysis is poorly constrained during these times.

## 6 Planetary Waves

In this section we investigate planetary waves in our radar wind observations over South Georgia. PWs are global-scale prop-agating waves with small zonal wavenumbers and periods of order days that arise as one of several rotational Hough modes in the Earth's atmosphere, where conservation of angular momentum is the restoring force that governs the oscillation (e.g. Smith, 2012). Note that we can only investigate travelling planetary waves using our single-site measurements here and not stationary planetary waves, which we are unable to separate from the large-scale flow.

To characterise the spectral properties of travelling PWs here, we use the $S$ transform (Stockwell et al., 1996). The $S$ transform is a spectral analysis technique that can provide time-frequency localisation of the complex spectrum, allowing us to probe the temporal variability of various periods present in our wind measurements. This is particularly useful for the study of PWs, whose amplitudes and periods may vary significantly over just a few cycles. The spectral coefficients of the $S$ transform are also directly related to wave amplitudes without the need for further scaling, which is an advantage over traditional forms of the continuous wavelet transform (CWT). The $S$ transform has also been used previously by Fritts et al. (2010b) to characterise PWs in meteor radar wind observations. Here we use the $S$ transform analysis of Hindley et al. (2019), which follows the same analytic approach as that of Stockwell et al. (1996) but includes several scaling options and significant improvements in computation speed. We select a scaling parameter of $c = 1$ to provide a fair balance between temporal and spectral localisation (see Hindley et al., 2019) meaning that any measured PW amplitudes can be considered to be "averaged" over approximately one wave cycle.

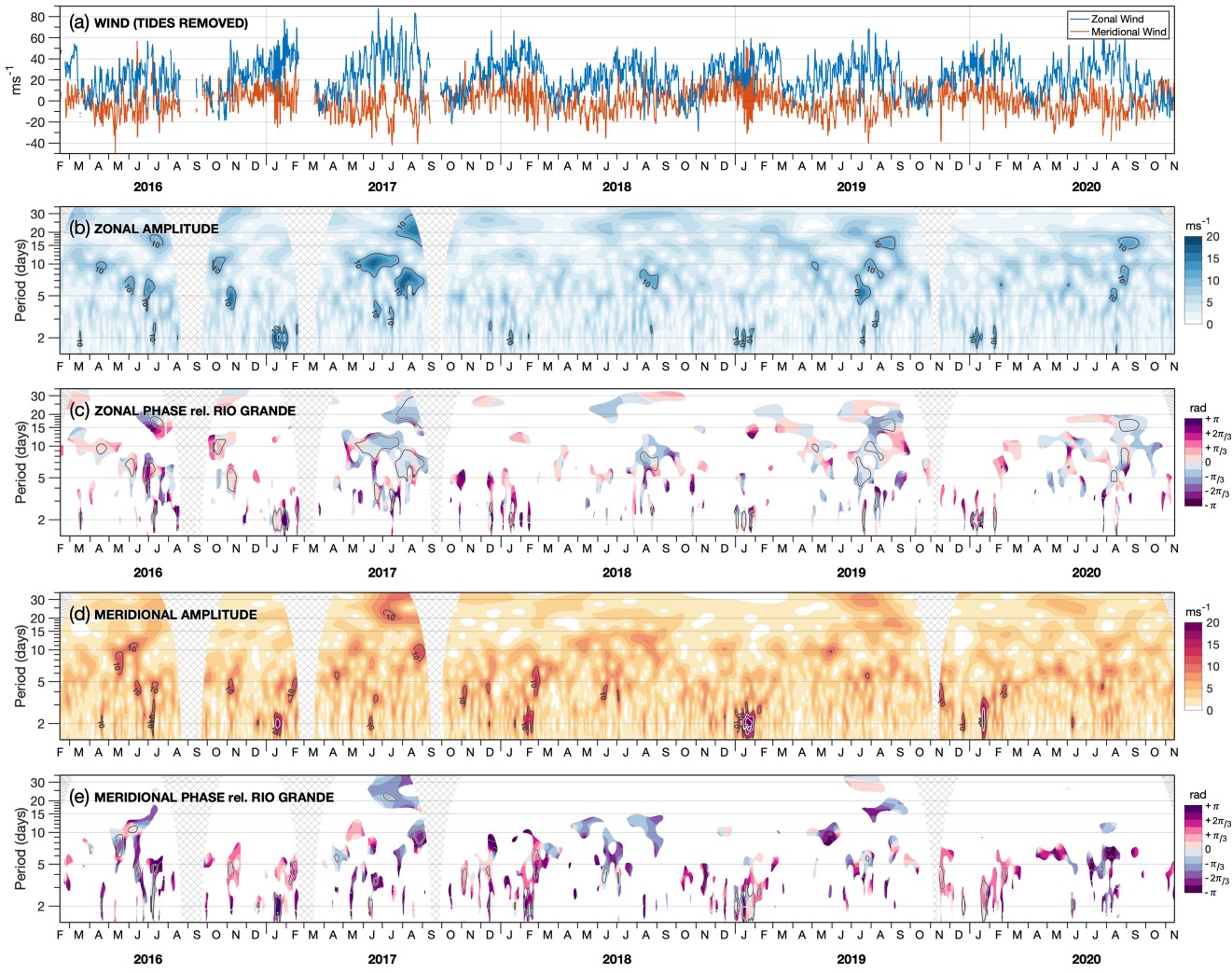

**Figure 6.** S-transform spectral analysis of planetary wave periods over South Georgia from meteor radar observations. Panel (a) shows 6-hourly zonal (blue) and meridional (orange) radar winds at 95 km altitude. Panels (b) and (d) show measured S-transform amplitudes of these zonal and meridional winds respectively, while panels (c) and (e) show the corresponding co-varying phase relative to 6-hourly wind measurements from the SAAMER meteor radar located 2000 km to the west. Here, a positive (negative) phase shift indicates a dominant eastward (westward) propagating PW mode. Hatched regions in (b,c) indicate unusable regions of spectral analyses due to missing data periods and "cone-of-influence" effects around these times.

Figure 6a shows 6-hourly zonal and meridional winds at 95 km altitude for the time period 2016 to 2020. These winds are found by subtracting a daily fit of the 24, 12, 8 and 6 hour solar tides as described in Sect. 5, then low-pass filtering the residual winds with a cut off of 6 hours. Figures 6(b,d) show PW amplitudes measured by our $S$ transform analysis for periods from



1.75 to 35 days. Hatched areas indicate missing time periods (due to the radar being offline) and unusable regions due to the "cone-of-influence" effect.

Periodic signals at known PW periods near approximately 2, 5, 10 and 16 days are observed. During December to February each year, a periodicity near 2 days is observed in both the zonal and meridional directions with amplitudes exceeding $10\,\mathrm{ms^{-1}}$. This is indicative of the Q2DW, which maximises during summer at high latitudes (e.g. Tunbridge and Mitchell, 2009). The largest measured amplitudes of the Q2DW seen here occur during January 2019, exceeding $15\,\mathrm{ms^{-1}}$ and $30\,\mathrm{ms^{-1}}$ in the zonal and meridional directions respectively. We found that the phase of the large-amplitude meridional Q2DW during this time (not shown in Fig. 6) is consistently around 0300 local time throughout January 2019. Interestingly, the phases of the meridional components of the diurnal and semidiurnal tides shown in Fig. 5 are also both around 0300 to 0600 local time, which could suggest a phase locking of the tides and the Q2DW that could lead to the large amplitudes observed.

Planetary wave activity at longer periods near 5, 6, 10 and 16 and 20 days is also observed in both the zonal and meridional directions in Figs. 6(b,d), with amplitudes greater than 10 ms persisting for more than a month on occasions. These periodicities likely correspond to presence of the quasi 5-day wave (Q5DW Day and Mitchell, 2010a), the quasi 6-day wave (Forbes and Zhang, 2017; Yamazaki et al., 2020), the quasi 10-day wave (Q10DW Forbes and Zhang, 2015; Wang et al., 2021) and the quasi 16-day wave (Day and Mitchell, 2010b). Each of these PWs typically reach a maximum at high latitudes during mid to late winter, which is consistent with each of the previous studies listed above. One interesting time period occurs during June to August 2017, where there is an apparent superposition of periods near 6-8 days, 10-11 days and 20-21 days in the zonal wind measurements, each up to $15\,\mathrm{ms^{-1}}$ in amplitude. It is likely that these periodicities correspond to the Q5DW, Q10DW and Q16DW respectively, but their periods may be modulated due to non-linear interactions with each other. In subsequent years 2019 and 2020, these PWs are measured again during June to August but with periods closer to their traditional periods near 5, 10 and 16 days, where they occur at slightly weaker amplitudes near to $10\,\mathrm{ms^{-1}}$. For these periods, the measured zonal wind PW amplitudes are almost always larger than the meridional, which is in contrast to the Q2DW amplitude during summer. We should note that there also exists a quasi 4-day wave (Yamazaki et al., 2021) that grows and maximises during summer at high latitudes, and this may be apparent during June 2017 in our results.

## 6.1 Eastward and westward propagating modes

Planetary waves observed at a single location can consist of a superposition of eastward (E) and westward (W) propagating modes, each with a zonal wavenumber e.g. 1, 2, 3. To investigate this we combine our results at KEP with wind measurements at the same altitude from the SAAMER meteor radar system at Rio Grande, in Tierra del Fuego (TDF), Argentina ($54°$S, $68°$W) to provide information on these eastward and westward propagating modes. We apply the same $S$ transform analysis to mean winds from TDF, then find the co-varying PW signals between the two sites as $C_{a,b} = S_a S_b^*$, where $S_a$ and $S_a$ are the $S$ transform spectra for TDF and KEP respectively, $C_{a,b}$ is the complex covariance spectrum and $S_b^*$ denotes the complex conjugate. We then find the phase difference of any co-varying signals between the two sites as $\tan^{-1}\left(\frac{\mathbb{R}(C_{a,b})}{\mathbb{I}(C_{a,b})}\right)$, where $\mathbb{R}$ and $\mathbb{I}$ denote the real and imaginary parts of $C_{a,b}$.





The results of this analysis are shown in Figs. 6(c,e) for the zonal and meridional direction respectively. Here, a positive (negative) phase shift indicates a dominant eastward (westward) propagating mode. The KEP and TDF radars are located approximately 2000 km apart, which is around $^1/_{12}$ of the circumference of the earth at this latitude ($\sim$23500 km). Therefore, measured phase differences near $\pi/6$, $\pi/3$ and $\pi/2$ correspond to dominant zonal wavenumbers of 1, 2 and 3 and so on. Regions where the measured PW amplitude at KEP is less than 5 ms$^{-1}$ are coloured white, and the 10 ms$^{-1}$ contour lines from panels b and d are added to indicate when PW activity is significant.

We find that the PW periods near 5, 10 and 16 days that occur at large amplitudes in the zonal direction during winter in Fig. 6b consistently correspond to small negative (blue) phase shifts in Fig. 6c, which indicates a dominant westward propagating mode for these PWs. For example, during June to September 2019 the measured phase shifts PWs near 5, 10 and 16 days are between $-\pi/6$ and $-\pi/3$, indicating dominant modes close to W1 and W2. The westward propagating Q10DW mode here is consistent with the results of Wang et al. (2021), who found evidence of a strong W1 mode of the Q10DW using meteor radar winds from Rothera and McMurdo, Antarctica during the months leading up to the 2019 southern sudden stratospheric warming (SSW Rao et al., 2020). During August to September 2020 in Fig. 6c, the small negative phase shift corresponding to the Q16DW is indicative of a dominant W1 mode, as predicted by Salby (1981b).

We should note that this analysis can only determine the dominant PW mode at any given time and period, and with only two sites we cannot unambiguously determine PW modes due to possible aliasing effects. However, because our two sites are relatively close together, this means that aliasing is unlikely for PW zonal wavenumbers smaller than perhaps 4 or 5. On the other hand, because the sites are so close, the phase shift for zonal wavenumber 1 is relatively small $\sim \pi/6$, which may be susceptible to measurement error.

The measured phase shifts for the Q2DW during summer each year in Figs. 6(d,e) also exhibit some unexpected results. Tunbridge et al. (2011) and Pancheva et al. (2018) used MLS temperature measurements to provide a global decomposition of the Q2DW into its eastward and westward propagating modes. They found that the large summertime amplitudes at high latitudes primarily correspond to the westward propagating zonal wavenumbers 2 and 3 (W2 and W3) modes. During winter, both studies also showed evidence of a weaker eastward propagating zonal wavenumber 2 (E2) mode of the Q2DW which, although maximising near the stratopause, was visible near 90 km. The MLS observations in Pancheva et al. (2018) were also supported by numerical simulations from the NOGAPS-ALPHA model, which supported the observed seasonal change in the dominant modes (Pancheva et al., 2016).

In our results however, we do not find a consistent phase shift during summer for the Q2DW. In the meridional direction, instead we measure a range of large positive phases shifts which would indicate eastward modes E3 and higher, which is not consistent with the studies mentioned above. Close inspection of the wind timeseries (not shown) indicates that these large positive phase shifts are clearly visible in the measurements. Further, the recent study of Fritts et al. (2021) found that although the W3 mode was dominant at this latitude during a large-amplitude Q2DW event, secondary modes of W1, W2, W4, E1 and E2 were also present. This suggests that we are unlikely to recover consistent phase information using only two sites, and more measurements around the circle of latitude are likely required to accurately characterise the dominant modes of the Q2DW.






**Figure 7.** Horizontal wind variance against height due gravity waves (GWs) for (a,b) large-scale "resolved" GWs in the derived winds and (b,c) small-scale "sub-volume" GWs. Panels (e,f) show the directional tendency of the sub-volume GW variance as described in Sect. 7. A positive (negative) directional tendency indicates a net zonal (meridional) preferential propagation direction. Bottom panels (g-j) show seasonal averages of zonal (black, bottom axis) and meridional (blue, top axis) winds against height from ERA5 reanalysis (solid) and the meteor radar (dotted) over South Georgia for the period 2016 to 2020.

# 7   Gravity waves

In this section we explore gravity wave activity over South Georgia in meteor radar observations. Gravity waves are a major contributor to the momentum forcing in the MLT that drives the residual circulation, including the cold summertime polar mesopause and much of the vertical transport of trace chemical species (e.g. Smith, 2012). Despite their importance, GWs are challenging to simulate in global models due to their small scales compared to model grid sizes, but their impacts can be global





scale (Alexander et al., 2010). Further understanding and quantification of the GW impacts in the MLT is needed to drive the development of the next generation of high-top numerical models that extend into the thermosphere.

### 7.0.1 Resolved and sub-volume GWs in radar measurements

There are two established ways to characterise GW activity in meteor radar observations, and we apply both methods here. The first method is to consider "resolved" GWs that can be resolved in the hourly derived winds over the whole radar collecting area (e.g. Song et al., 2017; Vargas et al., 2021). To do this, we subtract the fitted 24, 12, 8 and 6 hour tidal components and the daily mean from the zonal and meridional winds, and apply a low-pass filter to remove any remaining periods longer than 6 hours, leaving residual hourly wind perturbations $u_{res}$ and $v_{res}$ which we assume are due to GWs. These residual winds can 405 be further explored, for example using hodograph analysis as in the study of Song et al. (2017), but here we simply estimate the variance due to these resolved GW perturbations for comparison to the second method below. We derive variance due to resolved GWs by adding the resolved residual perturbations in quadrature $U_{res} = \left( u_{res}^2 + v_{res}^2 \right)^{1/2}$ and take the monthly variance of $U_{res}$ for each height.

One important consideration for this method is the careful removal of tidal components from the residual wind perturbations, 410 particularly the dominant semidiurnal tide at this latitude. Although GW periods can be up to $\sim$14 hours (the inertial period at 54°S), we found that simply removing the tidal components by fitting sinusoids still resulted in residual periodic features near 12 hours at large amplitudes that contaminated the variance residual perturbations due to GWs. For this reason we apply the 6-hour cut off filter to avoid this tidal contamination, and we assume that any higher order tides (4.8 h, 4 h etc.) are small. It should be noted however that residual periodicity near to tidal periods could be indicative of non-linear interaction between 415 tides and planetary waves (e.g. Beard et al., 1999), which can be investigated further in a future study.

The second method to characterise GW activity is to use the simple "sub-volume" variance method of Mitchell and Beldon (2009). For this method we take the measured radial meteor drift velocity for each meteor echo and subtract radial projection of the hourly horizontal zonal and meridional winds, which are interpolated to the time and height of each meteor position. These residual radial velocities of each meteor echo are assumed to be due to small-scale GWs that are not resolved in the 420 derived horizontal winds. We then take the monthly variance of these residual perturbations for each height.

One limitation of the sub-volume method is that it cannot distinguish between random errors in the radial velocity measurement or meteor geolocation (derived using interferometry between detections from the antenna array) and perturbations due to GWs. Over time, freeze-thaw weathering effects on the cables connecting the antennae array to the receiver can cause degradation of the signal, affecting the accuracy of meteor position determination. However, as has been shown in previous 425 studies (e.g. Mitchell and Beldon, 2009; Beldon and Mitchell, 2009), this approach exhibits a clear seasonal cycle that cannot be explained by systematic or random errors due to hardware degradation, and is supported by GW observations from satellite observations in the MLT (e.g. Liu et al., 2019).




### 7.0.2 The observational filter

The observational filter refers to the range of GW horizontal and vertical wavelengths and/or periods of GWs that the instrument is sensitive to (Preusse et al., 2002; Alexander and Barnet, 2007). The two methods for estimating GW variance applied here have mutually exclusive observational filters. That is, they are sensitive to different and non-overlapping parts of the GW spectrum.

The resolved GW variance method is sensitive to GWs with (i) periods $2 \lesssim \tau \lesssim 6\,$h and (ii) horizontal scales greater than around $400\,$km (the approximate horizontal collecting area of the radar) and (iii) vertical scales greater than around $3\,$km. These limits are determined by the specifications for the derived winds chosen in Sect. 3 and the 6-hour cut off filter. GWs detected by this method are likely to be inertia-gravity waves (IGWs) for which the effects of the earth's rotation are important (Fritts and Alexander, 2003).

The sub-volume GW variance method is sensitive to GWs that have (i) periods $\tau \lesssim 2\,$h or (ii) horizontal scales less than around $400\,$km or (iii) vertical scales less than around $3\,$km. Note that a GW only needs to satisfy *one* of these criteria to be detected (Davis, 2014). Note that this is quite different from the observational filters of, for example, satellite observations where a GW must satisfy *all* resolution limits to be resolved. We expect that this sub-volume method is predominantly sensitive to GWs that are relatively small-scale, that is, smaller than the collecting area of the radar which we expect to be the dominant factor of the three listed above.

### 7.0.3 Comparison of GW variances

Figure 7 shows monthly resolved (panels a,b) and sub-volume (panels c,d) GW variance against height over South Georgia for 2016 to 2020. As in Fig. 4, the rightmost panels (b,d) show an average composite year for the period 2016 to 2020. Note that values are around 4 to 5 times smaller in the resolved GW variance than in the sub-volume variance. This is expected because the sub-volume variance is of individual meteor drift velocities, which can vary much more significantly than the large-scale horizontal winds. It can also include some error in radial velocity and position.

Both GW variance methods exhibit annual and semiannual cycles in Fig. 7. Above $90\,$km altitude, a large wintertime maximum is observed and a smaller maximum is found during summer, with local minima near the spring and autumn equinoxes. At altitudes below $90\,$km, there is a summertime maximum with smaller values near the spring and autumn equinoxes in September and March and a weaker maximum during mid winter. These seasonal patterns are consistent with previous studies of high latitude GW variance in the MLT (Mitchell and Beldon, 2009; Song et al., 2021), but to our knowledge this is the first time the two GW variance methods have been compared using measurements from the same radar.

One interesting result is how closely the resolved GW variance follows a seasonal cycle, with a clear symmetry around the winter solstice in Fig. 7b. This is encouraging considering that the method is sensitive to only a small range of GW periods $2 \lesssim \tau \lesssim 6\,$h. The summertime maximum in the resolved variance is also proportionally larger than the equivalent summertime maximum in the sub-volume variance, suggesting a significant role for large scale IGWs in the MLT in summer.





The result that both methods exhibit similar seasonal activity suggests that GWs at a broad range of GW scales broadly follow the same seasonal pattern in the MLT at this location. There does not appear to be a significant period where one method shows large variance but the other does not, suggesting that the sources and filtering of these waves could follow similar patterns. It could also be the case that the small-scale GWs in the sub-volume method could be secondary GWs generated in situ from the breaking or dissipation of the larger scale GWs in the resolved method, as discussed by Vadas and Becker (2018). Although the modelling study of Becker and Vadas (2018) predicted scales of several 100s of km for these 2GWs, the recent studies of Lund et al. (2020) and Fritts et al. (2021) used a higher spatial resolution localised model over the southern Andes to suggest that 2GWs at much smaller scales of less than 100 km could be generated. Further exploration of this topic is needed to constrain the origins of the GW variance detected in meteor radar observations.

### 7.0.4 GW directions

Despite its simplicity, the sub-volume GW variance method also provides us with an opportunity to infer information about GW directions. Several previous studies have used the method of Hocking (2005) to derive directional GW momentum flux estimates from the meteor radar observations. However, this method has several important limitations that not discussed in the original manuscript, such as the assumption of either upward or downward GW propagation to break the directional ambiguity, circular polarisation of GWs and the omission of potentially important higher order terms in their analysis. Further, to our knowledge an experimental validation of momentum fluxes from this method is yet to be performed. For this reason, we do not apply the method here.

Instead, we apply a simple method to estimate directional information from the sub-volume GW variance. We apply an azimuthal weighting function to separately derive GW variance due to GWs that are aligned in the north-south or east-west direction. This method expands on the approach of Manson et al. (2004) who used "perturbation ovals" to investigate the azimuthal distribution of wind variance from meteor and medium-frequency (MF) radars. We can only apply this method to the sub-volume GW variance method because the directional information of individual meteor locations is lost when the resolved winds derived.

To split the residual radial wind perturbations into zonal and meridional directions, we define weighting functions in azimuth of $|\sin\theta|$ and $|\cos\theta|$ for the zonal and meridional directions respectively. We then compute the weighted variance of the residual radial wind perturbations $\sigma^2_{\mathrm{zon}}$ and $\sigma^2_{\mathrm{mer}}$ in the zonal and meridional directions using these weightings.

We then define the directional tendency $\tau$, which can be written as a percentage, as

$$\tau = 100 \times \left( \frac{\sigma^2_{\mathrm{zon}}}{\sigma^2_{\mathrm{mer}}} - 1 \right) \tag{4}$$

Here, a positive value for $\tau$ indicates that $\sigma^2_{\mathrm{zon}} > \sigma^2_{\mathrm{mer}}$ which suggests a zonal tendency in GW directions, and likewise a negative value for $\tau$ indicates a meridional tendency for GW directions.

The method cannot tell us if a GW is propagating eastward or westward, but it can tell us if the wave is orientated in the zonal or meridional direction. If a GW is propagating in the zonal direction, its phase fronts are likely to be aligned in the north-south plane. This means that any residual meteor wind perturbations due to this gravity wave measured to the east or west of the





radar will exhibit larger radial wind perturbations than those measured to the north or south. Therefore, if the variance of the residual perturbations in the zonal direction is larger than the variance in the meridional direction, we can infer that there is a zonal "directional tendency" of GWs towards the zonal direction. We can then use our knowledge of the local wind fields to further constrain whether these GWs are likely to be eastward or westward propagating.

Figure 7e shows the monthly directional tendencies for GW variance over South Georgia during 2016 to 2020. As before, the rightmost panel (f) shows the results for a composite year using all residual wind perturbations during all Januaries, all Februaries and so on during 2016 to 2020. Panels g to j show climatological zonal (black, bottom axis) and meridional (blue, top axis) winds averaged at each height level over a horizontal region 400 km radius from the island from ERA5 (Copernicus Climate Change Service, 2017) reanalysis, produced at the European Centre for Medium Range Weather Forecasts (ECMWF). Average winds are shown for four seasons during 2016 to 2020, and winds in the MLT from the South Georgia meteor radar are shown by the dotted lines.

A seasonal cycle in GW directional tendencies is apparent, with summertime GW variance around 20% larger in the zonal directional than in the meridional and a weaker tendency towards the meridional of around 10% during winter.

The positive GW directional tendency towards the zonal direction during summer in Fig. 7e coincides with westward summertime winds in the stratosphere below 90 km altitude in panel g. Indeed, the largest zonal tendencies coincide with the largest westward wind values during November 2018 to February 2019 in Fig. 4. We can infer therefore that the measured GW variance in the mesosphere over South Georgia during summer is likely to be due to eastward propagating GWs that are able to propagate freely up to the MLT and are likely to have a non-orographic origin, given the eastward winds near the surface. Any westward propagating GWs with sources in the troposphere (e.g. orographic "mountain" waves) are unlikely to reach the mesosphere during summer due to critical filtering by westward winds in the stratosphere and above.

However, the summertime mean winds in Fig. 4 also show a zonal wind reversal near 85 km altitude during summer, which would also filter out any eastward propagating GWs. The continuation with increasing altitude of the zonal tendencies during summer in Fig. 7 could be explained the impact of the solar tides and/or the Q2DW, which maximises during summer (see Fig. 6). This could strongly modulate the altitude at which the wind reversal occurs, allowing GWs to propagate vertically approximately half of the time. Note that GW breaking, large-scale turbulence and secondary GW generation are all expected to be detectable in the simple GW variance, not just propagating primary GWs. Therefore, it is not so unexpected to find some GW variance at altitudes above the zonal wind reversal.

A weak tendency towards the meridional direction during winter in Fig. 7 is an interesting result. It is expected from recent studies (e.g. Wright et al., 2017; Hindley et al., 2019, 2020, 2021) that the stratosphere exhibits a strong westward tendency in GW momentum flux over the Southern Ocean during winter for upwardly propagating GWs. If these westward GWs propagate up to the mesosphere, we would expect to see a strong zonal directional tendency in Fig. 7. But the fact that this is not observed suggests that GWs in the wintertime maximum in GW variance above 90 km altitude may have a variety of directions, or even a slight meridional tendency.

These results are consistent with recent theoretical, modelling and observational studies that discuss the important role of secondary GWs in the winter mesosphere near South Georgia (Vadas and Becker, 2018; Becker and Vadas, 2018; Vadas et al.,





2018; Kogure et al., 2020; Fritts et al., 2021). Westward orographic gravity waves may grow to such large amplitudes as they ascend that they become convectively unstable (where their peak-to-peak amplitudes exceed the dry adiabatic lapse rate) and
break, generating of 2GWs. These 2GWs may then themselves break, causing a further cascade of higher order 3GWs, 4GWs et cetera that propagate vertically.

Crucially, these 2GWs (and higher order) appear to consist of concentric rings in modelling and observational studies (Kogure et al., 2020; Lund et al., 2020; Fritts et al., 2021) that radiate in all directions, except for the direction exactly perpendicular to the original breaking GW. This could explain why we do not see a strong zonal tendency for GW variance during
winter above 85 km, as the GW directions are more evenly distributed in azimuth. Visual inspection of direct airglow observations of 2GWs in the MLT by Kogure et al. (2020) reveals a slight meridional tendency in concentric 2GWs measured near South Georgia, which would be consistent with the weak meridional tendency shown in Fig. 7.

In a recent numerical modelling study over South Georgia, Hindley et al. (2021) found that in some cases, mountain waves from South Georgia can reach altitudes near the stratopause at around 75 km if eastward wind conditions were favourable
throughout the stratosphere. In one example (their Sect. 8), the stratospheric portion of one modelled GW showed excellent agreement with co-located 3-D satellite observations. This could explain the smaller zonal tendency we find at altitudes below ∼80 km during winter in Fig. 7, which could indicate that some mountain waves may be able to propagate to this altitude before either becoming convectively unstable, or the strong tidal amplitudes form critical wind layers, if the wind conditions permit.

Although our results do not show direct evidence of 2GW activity, the occurrence of significant GW variance in the MLT during winter and the change in directional tendencies above 80 km altitude are consistent with expected 2GW predictions from recent theoretical and modelling studies. Our results therefore provide a strong observational inference of this 2GW activity in the wintertime polar mesosphere.

## 8   Comparison of radar winds and satellite temperatures to WACCM

In this section, we compare the observed winds from the South Georgia meteor radar and temperature measurements from MLS to climatological winds and temperatures in the Whole Atmosphere Community Climate Model (WACCM). WACCM is a comprehensive global climate model that extends from near the surface to the lower thermosphere, at around 140 km altitude. As global scale numerical climate models are being cautiously extended into the mesosphere and lower thermosphere, meteor radar measurements of wind remain one of the only continuous long-term wind measurement techniques that can be used to
constrain circulations and guide future development of these models. As discussed in Sect. 2.3, the WACCM simulations cover the period 1950 to 2014 and were prepared for the CMIP6 model intercomparison project.

Here we show the WACCM monthly-mean zonal and meridional winds and temperatures against height interpolated to the location of South Georgia. The selected run used an ensemble of three realisations from which we take the ensemble mean. We take a climatological average of WACCM monthly mean winds and temperatures for all years during 2000 to 2014
inclusive. This time period was carefully selected to be long enough for any oscillations near 11 years (e.g. solar cycle) to



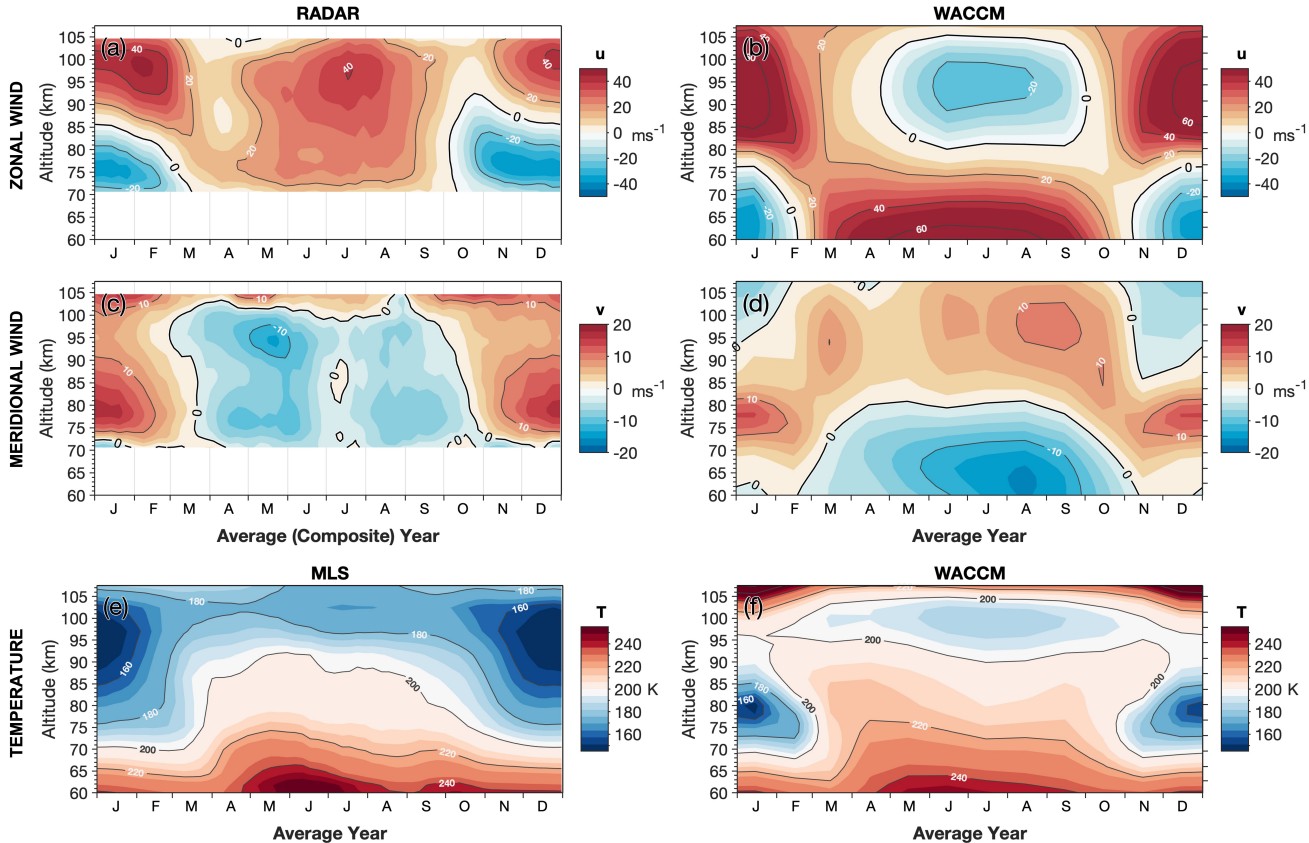

**Figure 8.** Comparison of an average year of meteor radar-derived zonal (top row) and meridional (middle row) winds and satellite-derived temperature measurements from MLS (bottom row) in the mesosphere and lower thermosphere to climatological winds and temperatures from WACCM simulations over South Georgia.

average out, but not so long that any changes due to long-term climate indices (e.g. $CO_2$) could have an impact (Ramesh et al., 2020). The WACCM monthly mean winds and temperatures in the MLT for this time period were carefully inspected (see supplementary Fig. S3), and interannual variability (such as the magnitude and sign of the zonal winds in winter or the height of the summertime zonal wind reversal) was found to be relatively small (that is, a few km or a few $ms^{-1}$). Therefore, we can

have good confidence that a meaningful comparison can be made between the climatological average of WACCM winds and temperatures for 2000 to 2014 and an average of meteor radar winds and MLS-derived temperatures over South Georgia for 2016 to 2020.





## 8.1 Comparison of zonal and meridional winds

Figure 8 shows average years of monthly mean meteor radar, WACCM and MLS winds and temperatures over South Georgia
from 60 to 105 km altitude. The MLS temperatures are averaged over a horizontal area less than 400 km radius from the island
for each height level, which is close to the size of the horizontal collecting area of the meteor radar. In panels b,d and f, tick
marks on the right hand axis indicate the approximate altitudes of the WACCM vertical grid levels.

During summer, both the radar and WACCM exhibit a realistic zonal wind reversal from westward to eastward in the MLT.
However, the wind reversal in WACCM occurs at around 75 km altitude compared to around 85 km in the radar observations,
and the magnitude of the summertime eastward jet is around 20 ms$^{-1}$ larger in WACCM. Further, the duration of these summer-
time conditions is shorter than observations indicate, lasting from only November to February in WACCM but from October
to March in the radar. Another interesting point is that the reversal of the observed zonal wind in the MLT at high latitudes
has a well-known characteristic "triangular" shape when plotted as in Fig. 4, with a rapid reversal from eastward to westward
at the onset of spring that slowly decays in height throughout summer into autumn (e.g. Fritts et al., 2010b; Sandford et al.,
2010). The zonal winds in WACCM do not appear to follow this characteristic structure in the wind reversal, instead showing
a smooth and equal transition in and out of summertime conditions.

The most significant difference however between the radar observations and WACCM in Fig. 8 is the direction of the
wintertime zonal winds. Radar and satellite observations indicate that the wintertime winds are eastward throughout the MLT
at high latitudes during winter, forming part of a mesospheric extension of the eastward stratospheric polar vortex (e.g. Beldon
and Mitchell, 2009; Harvey et al., 2019). In WACCM, the zonal winds decelerate and eventually reverse to westward above
85 km altitude throughout April to September. This was the case in every year of the WACCM run during 2000 to 2014 (see
supplementary Fig. S3) and is also the case for the 1850 to 2014 mean as shown by Ramesh et al. (2020). This difference is
well-known (e.g. Smith, 2012; Harvey et al., 2019) and may be considered as one of the most significant biases in numerical
simulations of the MLT with important impacts for MLT chemistry. There is a growing body of literature that suggests that this
reversal of the modelled zonal winds could be due to an incomplete representation of drag due to GWs, in particular secondary
GWs (Vadas and Becker, 2018; Becker and Vadas, 2018; Vadas et al., 2018). This aspect is discussed further in Sect. 9.

In the meridional direction, the magnitude and altitude of the largest northward winds during summer shows good agree-
ment between WACCM and the radar, reaching maxima of around 14 ms$^{-1}$ between 75 and 80 km. However, the summertime
northward flow in radar observations remains large and extends to 105 km altitude, while in the WACCM observations this flow
weakens and reverses to southward above 90 km altitude during November to February. During winter, flow is southward at
around 2 to 10 ms$^{-1}$ in the radar observations throughout the MLT, but in WACCM this flow weakens and reverses to northward
above 80 km, reaching northward values around 10 ms$^{-1}$ during winter that are not found in the radar. There is also an interest-
ing feature of a brief reversal from southward to northward in the radar winds during midwinter (around June-July) which is
found to occur in 3 of the 5 years in the radar observations (see Fig. 4) but is not seen in any year in WACCM during 2000 to
2014 (see supplementary Fig. S3).





## 8.2 Comparison of temperatures in the MLT

Temperatures from MLS and WACCM are shown in Figs. 8(e,f). The temperature structure of the MLT is one of the primary indicators of the residual circulation that we observe in our mean winds (Becker, 2012; Smith, 2012). Since both winds and temperatures affect wave propagation and dissipation, realistic simulations of both are necessary to adequately represent the
interactive dynamics.

The seasonal cycle is characterised by a cold summertime mesospause below 160 K and a warmer wintertime upper mesosphere of more than 200 K. The seasonal temperature variability in panels e and f of Fig. 8 closely corresponds to the wind patterns seen in panels a to d. In WACCM, the summertime mesopause is approximately the same temperature as observed by MLS, but it is tightly located to between 75 to 80 km altitude, which closely corresponds to the strong zonal wind gradient
with height in panel b and the largest northward winds in panel d. Conversely, the observed MLS temperature structure during summer has the same summertime mesopause temperatures below 160 K but it is centred at around 15 km higher in altitude and spread over a larger vertical region. This corresponds to a large vertical spread in the northward meridional wind in the radar in panel c and a smaller zonal wind gradient with height in panel a. As discussed above, the summertime temperature conditions in WACCM do not start as early or persist as long as the observations suggest. Cold polar mesopause temperatures have a first
order effect on the formation of polar mesospheric clouds, which are involved in the annual creation and destruction of key chemical species. An improvement in simulating these cold polar temperatures is therefore likely to lead to improved long-term forecasts of atmospheric chemistry at climate timescales in WACCM simulations that include these chemical processes.

During winter, the mesopause occurs at approximately the same altitude in both MLS and WACCM, but in WACCM it around 20 K warmer. This could be indicative of an anomalous westward GW drag in the polar region in WACCM that causes
an anomalous polar warming (Becker, 2012), driving a westward vertical shear of the residual wind with increasing altitude, which eventually reverses as shown in Fig. 8b. Further, a strong positive temperature gradient above the mesopause in WACCM indicates a faster transition into the thermosphere than observed by MLS throughout the year. This aspect could help to explain the reversal of the summertime meridional winds in WACCM from northward to southward above 90 km altitude and the lower vertical extent of northward winds during winter.

## 9 Discussion

### 9.1 Secondary GWs and differences between radar observations and WACCM

In Sect. 8 we highlighted several differences between the observed and simulated temperature structures and residual circulations in radar and satellite observations and WACCM. One key discrepancy is westward winds during winter in WACCM that are not observed by the radar.
Unlike the lower atmosphere, the temperature structure and residual circulation of the MLT region is highly sensitive to the drag and driving effects of atmospheric waves, particularly GWs (e.g. Smith, 2012; Becker, 2012). In global models however, parameterisations must be used to account for GW processes due to their relatively small size (particularly near their sources)





compared to the model grid (Holton, 1983; McLandress and Scinocca, 2005). The GW parameterisation in WACCM uses a discrete spectrum of GWs and is based on the approach of Lindzen (1981) with several updates (Richter et al., 2010; Garcia

et al., 2017).

Westward wintertime winds in the simulated polar MLT are a long-standing feature of WACCM (Harvey et al., 2019; Ramesh et al., 2020; Stober et al., 2021b), and they are also found in the thermospheric extension WACCM-X (Liu et al., 2010; Pedatella et al., 2014; Liu et al., 2018; Pancheva et al., 2020) and in other high-top models such as the MUAM (Lilienthal et al., 2018). Although not all global models currently show this feature, models that do simulate the observed eastward winds in winter may

each have other biases of their own in other altitude regions, as shown by Pedatella et al. (2014) and more recently by Stober et al. (2021b). Careful tuning of the GW parameterisation scheme in WACCM may likely produce a closer agreement to the observations for a given season, however this may give rise to other biases in different regions and seasons.

The long-standing nature of this discrepancy is indicative of the complexity of the processes involved, and likely the importance of accurately simulating the momentum deposition effects of GWs. As discussed by Becker and Vadas (2018) and Vadas

and Becker (2018), it has been proposed that at high latitudes during winter primary GWs propagate upwards from near the surface and undergo wave breaking and/or diffusion, which provides a local body force on the background flow. These local body forces then act to generate 2GWs, which they themselves break and generate tertiary waves and so on. This multi-step vertical coupling changes the direction of the momentum deposition in the MLT, since secondary waves are generated in concentric rings in all directions except for perpendicular to the direction of the initial primary wave (Vadas et al., 2018; Lund

et al., 2020; Fritts et al., 2021). This anomalous eastward drag causes and anomalous cooling over the polar cap, which results in an eastward wind shear with altitude in the residual circulation of the MLT as observed by the KEP radar in Fig. 8.

This means that even if the filtering of surface-launched parameterised GWs by the background winds is correctly specified in model parameterisations, and the columnar assumption of vertical wave propagation is reasonably valid, this final step of directional GW momentum deposition (either by primary or secondary GWs) must be accurately simulated to develop realistic

circulations in the MLT. The proposal of Vadas and Becker (2018) and Becker and Vadas (2018) is to resolve these 2GWs through the accurate simulation of local body forces on the atmosphere during wave breaking and dissipation, which then spontaneously generate propagating 2GWs. Another approach could be to develop a multi-height GW drag parameterisation scheme (e.g. Ribstein et al., 2020) which may be computationally cheaper to implement.

## 10   Summary and Conclusions

In this paper we have described a new SKiYMET meteor radar system deployed at King Edward Point on South Georgia island (54°S,34°W). The radar system made near-continuous measurements of winds in the mesosphere and lower thermosphere from February 2016 until it was removed in November 2020. Here we described a new Gaussian weighting method for providing improved time-height resolution and reliability of meteor radar winds and compared our measurements to a climatological WACCM simulation and MLS satellite temperature measurements. Our main results are:



1. We have characterised the zonal and meridional wind structure of the MLT at this remote location. We find an apparent semiannual cycle in the meridional winds, likely related to modulation of the upper branch of the Brewer-Dobson circulation by the mesospheric semiannual oscillation in the tropics.

2. We show seasonality of the diurnal, semidiurnal and terdiurnal solar tides in the MLT at this location. The semidiurnal tide is dominant and exhibits a strong annual and semiannual cycle, reaching $60\,\text{ms}^{-1}$ and $80\,\text{ms}^{-1}$ in the zonal and meridional directions respectively during April to May. The terdiurnal tide exhibits a similar seasonality to the semidiurnal tide and can have large amplitudes than the diurnal tide during winter.

3. We analysed for planetary wave periods near 2, 5, 6, 10 and 16 days and recovered their amplitudes and interannual variability. We find very large meridional wind amplitudes for the Q2DW during summer, where values exceed $30\,\text{ms}^{-1}$. Using wind measurements from the nearby SAAMER meteor radar at Rio Grande, we show that the dominant modes of the 5, 6, 10 and 16 day waves during winter are westward propagating modes W1 and W2.

4. We derived the horizontal wind variance due to large-scale "resolved" GWs and small-scale "sub-volume" GWs using two different methods, revealing a strong annual and semiannual cycle in GW activity. We then describe a new directional tendency method to quantify the directionality of this variance, showing a strong westward zonal tendency up to 20% during summer and a weak meridional tendency from 0 to 10% during winter. The fact that we do not observe a strong zonal tendency during winter, only in summer, is consistent with expected secondary GW generation and multi-step vertical coupling recently proposed to dominate the GW field in the wintertime MLT region at high latitudes.

5. We compared zonal and meridional winds from the radar and temperatures from MLS satellite measurements to climatological WACCM simulations from the CMIP6 project. We find that the WACCM simulations exhibit a summertime zonal wind shear that is stronger and ~10 km lower in altitude than observed, coincident with a simulated summertime mesopause that is the correct temperature (~160 K) but is too tightly localised in altitude and persists for around 2 months less than observed. There is also a reversal of the summer time meridional winds above 90 km that is not seen in observations.

6. We also found that during winter, the observed zonal winds above 80 km are eastward but in the WACCM simulation they are westward. We suspect that this is indicative of insufficient eastward momentum deposition by secondary GWs which occurs in the real atmosphere but cannot currently be simulated in existing parameterisations in WACCM, which leads to an unrealistic thermal structure and residual circulation in the MLT.

These results highlight the important contribution measurements of the MLT made at remote locations such as South Georgia can provide to develop our understanding of MLT wind, wave and tidal dynamics. These insights can be used to constrain and guide the development of general circulation models as they are extended into this dynamic region of the earth's atmosphere.

*Code availability.* All analysis and figure code will be made available in an online repository upon final publication.





*Data availability.* Meteor radar data from KEP are archived at the Centre for Environmental Data Archival (CEDA) and are freely available at https://doi.org/gjz3rr. MLS satellite data are available from NASA at https://disc.gsfc.nasa.gov/ and the WACCM modelling data used here are available from https://esgf-node.llnl.gov/projects/cmip6/.

*Financial support.* This work was supported by the UK Natural Environment Research Council (NERC) under grants NE/K015117/1, NE/K012614/1, NE/R001391/1, NE/R001235/1 and NE/S00985X/1 and the Royal Society under grant number UF160545.

*Author contributions.* The South Georgia meteor radar at KEP was installed by NJM and NC in 2016. It was supported by the SGWEX grant for which NJM and TMF were investigators, and later the DRAGON WEX grant for which NJM, TMF and CJW were investigators. The WACCM data were provided by AKS, and the SAAMER meteor radar data are provided by DCF and DJ. The radar, satellite and model data analysis, written manuscript and publication figures were produced by NPH, and all authors contributed to the final manuscript wording.

*Competing interests.* The authors declare that they have no competing interests.

*Acknowledgements.* We would like to thank the government of South Georgia and the South Sandwich Islands for their cooperation. In addition we would also like to thank the relevant staff at GENESIS, King Edward Point, British Antarctic Survey and University of Bath for all their help in ensuring the successful delivery of the instrument campaign. The SG-WEX project that deployed the radar was supported by the United Kingdom Natural Environment Research Council (NERC) under grants NE/K015117/1, NE/K012584/1, and NE/K012614/1. Its continuation was supported by the NERC DRAGON-WEX project under grants NE/R001391/1 and NE/R001235/1. The WACCM data used here derives from the CESM project, which is supported primarily by the United States National Science Foundation (NSF) and is based upon work supported by the National Center for Atmospheric Research (NCAR) sponsored by the NSF under Cooperative Agreement No. 1852977. Computing and data storage resources, including the Cheyenne supercomputer (doi:10.5065/D6RX99HX) were provided by the Computational and Information Systems Laboratory (CISL) at NCAR.



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
