# Peer review of "Radar observations of winds, waves and tides in the mesosphere and lower thermosphere over South Georgia island (54°S, 36°W) and comparison to WACCM simulations"

_Atmospheric Chemistry and Physics, 2021_

## Referee Comment (RC1)

**Review of:**
Radar observations of winds, waves and tides in the mesosphere and lower thermosphere over South Georgia Island (54°S, 36°W) and comparison to WACCM simulations.
Manuscript ACP https://doi.org/0.5194/acp-2021-981
by Neil P. Hindley et al.

It is always good to have data from a new location. There is a very impressive list of references ( it is a bit hard to tell which are the most important). An important comment made is that different models may be better at some heights and locations than others - a reminder of something which is easily forgotten.

Figure 1: given the "lopsided" azimuthal distribution, please add a comment on the possible, or not, effects on the analysis results.

Comments on Section 3.1: inversion of an NxN matrix is not necessary.

$$e_i^2 = \left[ \frac{V_{r_i}}{\sin \phi_i} - W_i(V_N \cos \theta_i + V_E \sin \theta_i) \right]^2$$

$$\frac{\partial \Sigma e_i^2}{\partial V_N} = 0$$

$$\frac{\partial \Sigma e_i^2}{\partial V_E} = 0$$

That is two equations in two unknowns.

I would like to argue about the Fig 3. significance levels which based on a Lomb-Scargle-type analysis. The original L-S papers assume that variance is divided equally between frequency bins- that is the spectrum is flat-ish. Since geophysical spectra are generally pink noise, the variance in the low frequency bins, noise or not, is already higher than the equally divided variance- so it is *easier* for those frequencies to be *significant.*
However those are still accepted in journals .... so I guess I can't complain

Pg. 7. A further refinement could be applied: because the meteor rate has a strong peak (around 90 Km), actual heights are effectively nearer the peak because increased number weighting. A way around this is to un-weight the echoes according to the local meteor rate so each height has equal weight, e.g. by fitting a Gaussian to the average height profiles, and weighting each meteor, "i", by it inverse, $(1/W_1)(i)$, the wanted Gaussian height profile, $W_2(i)$, and wanted time profile $(W_3)(i)$.

Pg. 10 line 400-406. Please give more details. At the moment it reads as if all the tidal removal and high-pass filtering operations are performed on a single day (24 hrs) interval(s), and if they are, then the 12 hr was removed. If such residual days are stuck together, considering

that the residuals may not have zero daily mean, the discontinuities between days may create 24, 12hr,... artifacts. Also, were the tides and mean removed in one fit, or sequentially?

Pg. 10. Line 414-415. The most likely reason for residual ∼12hr periods is sidelobes of a varying tide. A sine wave over a long time interval has a narrow bandwidth - a notch filter when subtracted, so some sidelobes may not be removed. Please add a comment, or modify the discussion accordingly.

Pg. 18. Comments on "sub variance" lines 416-427: Because of radial echo measurement, it is not possible to get independent component variances (e.g. North, East). E.g. perturbations in eastward wind will "bleed" into the northward by an amount depending on echo azimuths. A simple model quickly shows the resulting re-distribution of variance. The actual distribution of echo azimuths adds to the problem. Selection of just ∼N and ∼E echoes might help - but in practice it results in very noisy wind values, because there are never enough echoes. For this reason conclusions based on the "sub-volume" component variances should be avoided. On the other hand, the total variance is a meaningful parameter; that is rms (radial velocity minus radial velocity from fit).

**minor, and typos**

Fig 2 caption: "(a,c)" and "(b,d)"

Pg. 3. line 79- "It operates as an interferometer ..."

Pg 3. line 84 "radial component of drift velocities ..." [ but I don't agree with this unless vertical velocity is assumed zero]

Pg 4. line 90 "receivers are blanked during each ..."

Pg 4. line 91 "to avoid saturating the receivers"

Line 115: "greater than ∼10 km ..." according to V5 documentation.

Pg. 11, Line 240: "The reason for this ..." What is this? the low meteor count?

—///—

---

## Referee Comment (RC3)

**Review of "Radar observations of winds, waves and tides in the mesosphere and lower thermosphere over South Georgia island (54°S, 36°W) and comparison to WACCM simulations"**

**General comments**

- This manuscript presents comprehensive analysis of horizontal winds observed using the meteor radar deployed at South Georgia island over the Southern Ocean and temperature obtained from the MLS sensor mounted on the AURA satellite in terms of mean flow, tides, planetary waves and gravity waves. In addition, this study compares the analysis results with the climatological WACCM results and suggest directions in which whole atmosphere climate models like WACCM can be improved in the future for better MLT simulations.

- The conventional methodology in retrieving horizontal winds from meteor radar echoes is overcome through the development of a novel technique based on Gaussian weighting approach. This new technique allow for obtaining the smoother wind information as much as possible from meteor radar observations. This study is impressive in that it includes almost every analysis that can be done with meteor radar horizontal winds. Most of descriptions of results are reasonable, but there is an issue that need to be clarified. This issue is related to conclusion of this study and may affect considerably discussions about secondary gravity waves. For this reason, reviewer recommends major revision for consideration of publication to Atmospheric Chemistry and Physics.

**Major comments**

- One major issue is about inferring the propagation direction of gravity waves derived from the sub-volume gravity-wave variances. Authors said in the paragraph between L490 and L496 that any residual meteor wind perturbation due to this (zonally propagating) gravity wave measured to the east or west of the radar will exhibit larger radial wind perturbations than those measured to the north or south. Therefore, if the variance of the residual perturbations in the zonal direction is larger than the variance in the meridional direction, we can infer that there is a zonal "directional tendency" of GWs towards the zonal direction.

  - First of all, could you confirm the formulations of $\sigma_{zon}^2$ and $\sigma_{mer}^2$? Are they defined by something like the following equations? $\sigma_{zon}^2 = \left( \sum_{i=1}^{N} v'^2_{ri} \sin^2 \theta_i \right) / (N-1)$ and $\sigma_{mer}^2 = \left( \sum_{i=1}^{N} v'^2_{ri} \cos^2 \theta_i \right) / (N-1)$, where $v'_{ri}$ is the residual radial (horizontal) wind at each radar echo position, $\theta_i$ is the azimuth angle of the radial (horizontal) wind measured clockwise from the North. Here, the residual wind is the result of subtracting radial projection of the hourly zonal and meridional winds from the observed radial winds on the horizontal plane as in

Mitchell and Beldon (2009).

– If the two formulations are what the authors employed, the directionality obtained based on these equations may give biased estimate of the propagation direction of gravity waves because $\sin^2 \theta_i$ ($\cos^2 \theta_i$) have the same positive weights for eastward and westward (northward and southward) propagating gravity waves. Figures below show possible situations of the phase progressions for gravity waves around a radar site. On the first panel, upward propagating gravity waves propagate eastward (or northward) relative to the zonal (meridional) mean flow. Radar observation is sensitive to those gravity waves east (north) of the radar, but not sensitive to those waves west (south) of the radar. On the second panel, upward propagating gravity waves propagate westward (southward) relative to the mean flow. Radar observation is sensitive to those gravity waves west (south) of the radar, but not sensitive to waves east (north) of the radar. In other words, radar observations can be sensitive to gravity waves that are generated from the radar site and propagate in radial directions away from the radar. However, for gravity waves passing through the radar site, radar can detect those waves only either east or west (either north or south) of the radar site. In summary, the directionality computed in the manuscript can possibly be biased towards gravity waves generated near the radar site (or downward propagating toward the radar site in case of upward phase propagation).

– Reviewer wonders if the authors have considered the above-mentioned issue. If the directionality computed is biased towards gravity waves propagating horizontally outward from (or towards) the radar site, the directionality in the present manuscript should be interpreted differently. Correction of the interpretation of the directionality may lead to modification of interpretation of the seasonal variations of the propagation direction of gravity waves and modification of discussion of generation of secondary gravity waves.

**Specific comments**

- L11: Symbols W1 and W2 are used without appropriate definitions.

- L41: Solomon and Garcia, 1987

- L52: Considering oblique propagation of GWs in parameterizations has a relatively long history, although Kalisch et al. (2014) is relatively recent one. Consider citing some of relevant previous studies: Song and Chun (2008), Hasha et al. (2008), or Choi et al. (2009).

- L53: Current GW parameterizations do neither include secondary GWs nor GWs that originate from the middle atmosphere (upper stratosphere or lower mesosphere) (e.g., Fairlie et al., 1990; Sato and Yoshiki, 2008).

- L220: How large-amplitude Q6DW can be related to small impacts due to southern SSW in September 2019?

- L222: It does not seem reasonable to discuss the Brewer-Dobson circulation (BDC) at the mesopause heights, although the BDC is expected to manage to reach somehow middle mesosphere as inferred from the stream lines shown in Seviour et al. (2012). Besides, results in Murgatroyd and Singleton (1961) were mostly shown below $z = 80$ km.

- Figure 5: How about include labels for zonal and meridional winds in panels (or as panel titles)

- L301: How can we explain the upward propagation of diurnal tides in the high-latitude regions where the Coriolis frequency is larger than diurnal frequency?

- L399: The subsection number should begin from 1.

- L465: 100s $\Rightarrow$ hundreds

- L532$-$L537: Secondary gravity waves with concentric-ring shaped phase lines can have meridional components of horizontal propagation even though they are generated by a zonal body force. That being said, logic used to explain the meridional GW propagation tendency seems weak because the major propagation direction of secondary GWs still lies in the sense of the direction of the body force induced by the primary GWs. This part is also related to the major issue raised above.

- L535: stratopause $\Rightarrow$ mesopause

- L594: WACCM observations $\Rightarrow$ WACCM

- Section 9: Although authors suggest impacts of secondary GWs as a promising mechanism to improve the MLT simulations of whole atmosphere models based on recent studies (e.g.,Becker and Vadas, 2018), there are also substantial uncertainties regarding impacts of GW generation due to flow imbalance in the upper stratosphere and lower mesosphere. Recommend to mention even briefly potential middle-atmospheric GW sources as mentioned above.

**References, not cited in the manuscript**

- Song, I.-S., and H.-Y. Chun, 2008: A Lagrangian spectral parameterization of gravity wave drag induced by cumulus convection. Journal of the Atmospheric Sciences, 65, 1204$-$1224. https://doi.org/10.1175/2007JAS2 369.1

- Hasha, A., O. Buhler, and J. Scinocca, 2008: Gravity wave refraction by three-dimensionally varying winds and global transport of angular momentum. Journal of the Atmospheric Sciences, 65, 2892$-$2906. https://doi.org/10.1175/2007JAS2561.1

- Choi, H.-J., H.-Y. Chun, and I.-S. Song, 2009: Gravity wave temperature variance calculated using the ray-based spectral parameterization of convective gravity waves and its comparison with Microwave Limb Sounder observations. Journal of Geophysical Research: Atmospheres, 114, D0811. https://doi.org/10.1029/2008JD011330

- Fairlie, T. D. A., M. Fisher, and A. O'Neill, 1990: The development of narrow baroclinic zones and other small-scale structure in the stratosphere during simulated major warmings. Quarterly Journal of Royal Meteorological Society, 116, 287−315. https://doi.org/10.1002/qj.49711649204

- Sato, K., and M. Yoshiki, 2008: Gravity wave generation around the polar vortex in the stratosphere revealed by 3-hourly radiosonde observations at Syowa Station. Journal of the Atmospheric Sciences, 65, 3719−3735. https://doi.org/10.1175/2008JAS2539.1

- Seviour, W. J. M., N. Butchart, and S. C. Hardiman, 2012: The Brewer-Dobson circulation inferred from ERA-Interim. Quarterly Journal of Royal Meteorological Society, 138, 878−888. https://doi.org/10.1002/qj.966

---

## Author Comment (AC1)

**Authors' response reviewers' comments on acp-2021-981: "Radar observations of winds, waves and tides in the mesosphere and lower thermosphere over South Georgia Island (54°S, 36°W) and comparison to WACCM simulations."**

N. P. Hindley et al.

We would like to thank the reviewers for their time and careful consideration of our submission, and their very helpful comments all of which have led to an improved manuscript. We have applied all of the suggested minor revisions from the reviewers, and removed the analysis and results derived from the new directional gravity wave tendency method pending further investigations of potential sources of error, as suggested in a major comment by Reviewer #3.

Please find our itemised responses to the reviewers individual points below.

**Response to Reviewer #1**

*"It is always good to have data from a new location. There is a very impressive list of references ( it is a bit hard to tell which are the most important). An important comment made is that different models may be better at some heights and locations than others - a reminder of something which is easily forgotten."*

Thank you for the comment, this was indeed one of the motivations for comparing the radar observations to the latest numerical models. Models are routinely compared between each other but it is important to compare to observations at new locations to explore any deviations from reality.

**Specific Comments**

1. **Figure 1:** *given the "lopsided" azimuthal distribution, please add a comment on the possible, or not, effects on the analysis results.*

   We assume the reviewer is referring to the reduction is observed meteor echoes in the outer "ring" of Fig. 1a between azimuths of around 345 to 60 degrees, which is caused by the nearby orography obscuring a small region of the sky at high zenith angles. These meteor echoes actually correspond almost entirely to high zenith angles (greater than 65 degrees) which are removed (l.88) before the wind fitting is performed (see Fig. 1c), so they do not affect our results.

   We neglected to include a histogram of meteor azimuths in the paper, so we have included one here in our public response for completeness (Fig. R1). The increased number of meteor counts to the south are expected at this latitude due to the geometry of the prevailing ionised meteor trails. This is because radar's detection of these trails works best when the trails are tangential to the radial direction, which occurs more often to trails on the poleward side of the radar. Meteor radar systems near the equator by comparison have an almost complete uniform distribution of meteor detections with azimuth (e.g. Davis et al., 2013). Either way, because meteors from all azimuths are included in the wind fitting, and the cosine function is unique at all locations within one period, an asymmetric distribution of meteor counts with azimuth does not affect our wind results. We have updated the text to reflect this.

2. **Comments on Section 3.1:** *inversion of an NxN matrix is not necessary... That is two equations in two unknowns.*

Agreed, we have removed this sentence and revised text as it was confusing. The matrix inversion is simply the method we use to solve Eqn. 3, which can be much more elegantly written if $W$ is $N$x$N$, but this is not efficient to solve computationally because the matrix would be much larger than is necessary, hence the comment.

3. *I would like to argue about the Fig 3. significance levels which based on a Lomb-Scargle-type analysis. The original L-S papers assume that variance is divided equally between frequency bins- that is the spectrum is flat-ish. Since geophysical spectra are generally pink noise, the variance in the low frequency bins, noise or not, is already higher than the equally divided variance- so it is easier for those frequencies to be significant. However those are still accepted in journals... so I guess I can't complain*

We completely agree, and we only included the Lomb-Scargle significance level to be consistent with these other studies. Ideally we would have a spectrally localised significance level that varied with the power of different parts of the underlying atmospheric frequency spectrum - but that would require a priori knowledge of the whole spectrum. We have added a sentence to clarify that this significance level is assuming a flat spectrum, and the height of a peak around its local neighbours is probably a better indication of significance for atmospheric spectra - but this is difficult to quantify.

4. **Pg. 7** *A further refinement could be applied: because the meteor rate has a strong peak (around 90 Km), actual heights are effectively nearer the peak because increased number weighting. A way around this is to un-weight the echoes according to the local meteor rate so each height has equal weight, e.g. by fitting a Gaussian to the average height profiles, and weighting each meteor,"i", by it inverse, (1/W1)(i), the wanted Gaussian height profile, W2(i), and wanted time profile (W3)(i).*

This is a good suggestion, and this is actually the first approach we tried. However, we found that this approach results in uncomfortably large weightings for meteors at high and low altitudes far away from where the peak of the Gaussian was centred, because of the division by the inverse Gaussian function.

What we do instead (see l.165-168) is take the weighted mean of the times and altitudes of all the meteors used for the weighted fit and use this as the time-height location of each derived wind measurement. All subsequent results figures in the paper are plotted using these weighted locations. The time and height offsets between the centre of the Gaussian functions and the weighted mean of the meteor echo locations is now included in the revised Fig. 2.

5. **Pg. 10 line 400-406:** *Please give more details. At the moment it reads as if all the tidal removal and high-pass filtering operations are performed on a single day (24 hrs) interval(s), and if they are, then the 12 hr was removed. If such residual days are stuck together, considering that the residuals may not have zero daily mean, the discontinuities between days may create 24, 12hr,... artifacts. Also, were the tides and mean removed in one fit, or sequentially?*

We assume the reviewer is referring to Sect. 5 of the submitted manuscript. We agree that the text here is confusing, especially the second and third paragraphs of Sect. 5, so we have rewritten the text to make it clearer how the tidal fitting was done.

We fit tides by first making a composite day of hourly zonal and meridional winds against height derived using measurements in a 30-day time window centred on this date, where all echoes in the time window are assumed to occur on the same day.

We then fit sinusoids of 24, 12, 8 and 6 hours to these hourly winds simultaneously. This is done by assuming that the daily hourly winds for each height consist of a sum of four sinusoidal waves at tidal periods $n$ described by constants $A_n$ and $B_n$ oscillating around a mean background wind $C$. For the hourly zonal wind $u(t)$ at one height, this can be written as

$$u(t) = \begin{pmatrix} \boldsymbol{A} & \boldsymbol{B} & C \end{pmatrix} \begin{pmatrix} \cos(2\pi t/\boldsymbol{\tau}) \\ \sin(2\pi t/\boldsymbol{\tau}) \\ 1 \end{pmatrix} \tag{1}$$

where $t$ is a vector of the weighted average times of the wind measurements in hours, $\boldsymbol{A} = (A_{24}, A_{12}, A_8, A_6)$ and $\boldsymbol{B} = (B_{24}, B_{12}, B_8, B_6)$ are row vectors containing the tidal amplitude coefficients for the cosine and sine terms respectively, $C$ is the mean and $\boldsymbol{\tau} = (24, 12, 8, 6)$ is a column vector containing tidal periods in hours. This can be arranged as $P = Mx$, which we solve for $x$ as $x = M^{-1}P$ using a matrix inversion, just like the wind fit in Eqn. 1, giving us the tidal coefficients. Tidal amplitudes and phases for each tidal period $n = 24, 12, 8, 6$ are given by $\sqrt{A_n^2 + B_n^2}$ and $\tan^{-1}(A_n/B_n)$ respectively, and $C$ is the mean background wind for the day. We then repeat this process to fit meridional tidal coefficients to the composite day meridional winds $v(t)$.

This revised description has been included in the revised manuscript.

The reviewer may correct us, but we believe this process is mathematically equivalent to first fitting the 24h period, subtracting it, then fitting the 12h period etc, because there is no filtering involved, it is just linear addition and subtraction.

6. **Pg. 10. Line 414-415:** *The most likely reason for residual $\sim$12hr periods is sidelobes of a varying tide. A sine wave over a long time interval has a narrow bandwidth - a notch filter when subtracted, so some sidelobes may not be removed. Please add a comment, or modify the discussion accordingly.*

We completely agree. In order for the semidiurnal tide to change phase throughout the year, it must have an apparent period that is very close to, but not perfectly 12 hours, hence the sidelobes and the imperfect removal by sine-fitting, and this is before even considering the effects of measurement error. We have experimented with various different methods, including notch filtering, to try and overcome this but there doesn't seem to be a more effective way than the rather heavy handed sine fitting, which has the advantage of coping well with data gaps. We have revised the text to mention this issue.

7. **Pg. 18. lines 416-427:** *Comments on "sub variance": Because of radial echo measurement, it is not possible to get independent component variances (e.g. North, East). E.g. perturbations in eastward wind will "bleed" into the northward by an amount depending on echo azimuths. A simple model quickly shows the resulting re-distribution of variance. The actual distribution of echo azimuths adds to the problem. Selection of just $\sim$N and $\sim$E echoes might help - but in practice it results in very noisy wind values, because there are never enough echoes. For this reason conclusions based on the "sub-volume" component variances should be avoided. On the other hand, the total variance is a meaningful parameter; that is rms (radial velocity minus radial velocity from fit).*

We have removed the directional tendency results from the revised paper in line with suggestions from Reviewer #3 (see below). We decided that more testing needed to be done to ensure that errors due to meteor height finding were not introducing spurious variance results in different directions.

For completeness, we will respond to the reviewer's points on this, although the section has been removed. We agree with the issues that the reviewer has raised, however our weighted method as described does overcome each of these points to provide a meaningful estimate of variance in the zonal and meridional directions:

*"it is not possible to get independent component variances (e.g. North, East). E.g. perturbations in eastward wind will "bleed" into the northward by an amount depending on echo azimuths"*: we are aware of this, which is why we use a weighted variance calculation where the variance is weighted towards echoes to the north/south or east/west (see text below for a better description of the weighting method, thank you for raising this). This is **not** the same as projecting the radial velocities into zonal/meridional directions and then taking the variance of each, which we agree would be meaningless due to the aforementioned bleed.

Next, *"Selection of just N and E echoes might help..."*: this is exactly what our weighting method does - it selects North/South and East/West echoes, but in a more elegant method than simply setting azimuth limits.

*"...but in practice it results in very noisy wind values, ..."*: We do not need to fit winds with these echoes - the mean wind fitting is already done as described in Sect. 3, using all available azimuths, and then these are subtracted to provide the residual radial perturbations. We then apply our azimuth weighting to these residual perturbations and compute the weighted variance.

*"...because there are never enough echoes"*: We compute the weighted variance for a sliding 30-day time window, which results in variance calculations involving several 1000s of meteor echoes. The mean and standard deviation are very well defined for such a distribution.

*"For this reason conclusions based on the "sub-volume" component variances should be avoided."*: We agree with the reviewer here, and although our directional results show some broad similarities with the "perturbation oval" analysis of Manson et al. (2004, their Fig. 10), we have removed them from the revised manuscript pending a more thorough assessment of potential sources of error in the method (see response to Reviewer #3 below for details).

We have also included a brief mention of other recent developments in sophisticated methods for deriving GW properties from radar datasets (e.g. Stober et al., 2021a) that may be applied to the South Georgia measurements in future.

**Minor Comments and Typos**

- *Fig 2 caption: "(a,c)" and "(b,d)"*

  Fixed, thanks.

- *Pg. 3. line 79- "It operates as an interferometer ..."*

  Fixed, thanks.

- *Pg 3. line 84 "radial component of drift velocities ..." [ but I don't agree with this unless vertical velocity is assumed zero]*

  Fixed, thanks. Yes, we usually assume vertical velocity to be zero because the horizontal velocity is normally so much larger, but agreed the measured value is the radial component of the measured drift velocity.

- *Pg 4. line 90 "receivers are blanked during each ..."*

  Fixed, thanks.

- *Pg 4. line 91 "to avoid saturating the receivers"*

  Fixed, thanks.

- *Line 115: "greater than ∼10 km ..." according to V5 documentation.*

  Fixed, thanks.

- *Pg. 11, Line 240: "The reason for this ..." What is this? the low meteor count?*

  We meant the reason for the unusual wind conditions observed in 2016 compared to later years, we have revised the text to make this clearer.

[Figure]

Figure R1: Histogram showing of average number of meteor echo detections per day against azimuth from the South Georgia radar for the period 2016 to 2020. Blue (grey) bars show meteor echoes detected between (outside) zenith angle limits of $15°$ and $65°$ from the zenith.

**Response to Reviewer #2**

*"The paper presents results of meteor radar measurements of mesosphere/lower thermosphere (MLT) winds and waves at a remote site at South Georgia island. The MLT is a crucial region of the atmosphere, with wave-mean flow interaction as a major mechanism that drives the background circulation. Detailed high-resolution continuous observations especially in the Southern Hemisphere are therefore very useful for understanding atmospheric dynamics and validating numerical models. The paper begins with the radar description and the presentation of a new method of data analysis, which may allow to extend the vertical range of measurements. Results for mean winds, tides, planetary waves and gravity waves are shown. The paper also includes a comparison of the results with WACCM circulation model predictions, indicating some differences and also partly known issue with numerical modeling of MLT dynamics. The paper is well written, the presentation is clear and the conclusions are well supported by the observations. I recommend publication after considering some minor comments below."*

We would like to thank the reviewer for their time and effort in reviewing the manuscript. We gladly apply all of their suggested points below.

**Specific Comments**

1. **Throughout** *The radar is sometimes referred to as "South Georgia radar" and as "KEP radar", which is a bit confusing*

   Thanks, we have fixed this.

2. **l.76-83** *L 76-83: please add peak power*

   Peak power was around 7.5 kW. We have added this, thanks.

3. **Figures 2 and 4** *daily data gaps above 100 km seem to match the poleward maximum of the meridional wind. Could this lead to an equatorward bias? There are rather strong equatorward meridional winds in summer at that height*

*region as shown in Fig 4. But there is some contradicting evidence for poleward meridional winds above 100 km (e.g. Qian et al., 2017). Could you please comment on that?*

This is an interesting point. In the study of Qian et al. (2017), SD-WACCM simulations show a reversal from equatorward to poleward meridional winds at high midlatitudes between altitudes around 91-106km during summer. They compare $CO_2$ distributions from WACCM to SABER observations that provided at least some validation of the simulated circulation.

However, as the reviewer points out, our local radar observations over South Georgia do not show any summertime meridional wind reversal between these altitudes as predicted by SD-WACCM. This discrepancy is also highlighted in our Fig. 8, which shows a similar meridional wind structure in SD-WACCM to the Qian et al. (2017) study.

Our results are however consistent with observations from several other meteor radar sites around the world as recently highlighted in Stober et al. (2021b). Of the six radar sites investigated by Stober et al., including four high latitude locations in Argentina, Antarctica and Scandinavia, none of these observations showed a significant summertime meridional wind reversal from equatorward to poleward flow between altitudes of 80-100km as predicted by SD-WACCM. Observations from the more powerful radar system at Rio Grande in Tierra del Fuego, Argentina (also used in this study), which is the closest location to South Georgia, extended up to 110km altitude but also did not show a meridional wind reversal. Further, radar observations near the equator also do not show any reversal between 82 and 95km as reported by Davis et al. (2013, their Fig. 4). This information has been included in the revised manuscript.

We don't doubt that such a circulation could exist in the lower thermosphere, as has been discussed by Qian et al. (2017) and Liu (2007), but these radar observations indicate that it could perhaps take place outside of this height range, where the radars cannot measure. Further study from a more global set of observations may be required to resolve this issue, but this is beyond the scope of the current study and we cannot speculate further here.

The reviewer also noted the daily data gaps in our Fig. 2:

*"daily data gaps above 100 km seem to match the poleward maximum of the meridional wind. Could this lead to an equatorward bias?"*

We have high confidence that the summertime equatorward flow at high altitudes is not due to a measurement bias caused by this and, as mentioned above, our results are consistent with other radar studies.

To explain, these data gaps in Fig. 2 are caused when there are too few meteor echoes during local time afternoon to derive a reliable wind fit at altitudes far from the peak meteor altitude near 90km (see Fig. 1b,e). If one were to take a simple daily average of these data, there could indeed be a bias caused by an incomplete cycle of the tide, as the reviewer correctly points out.

However in the analysis used for Fig. 4, we overcome this problem by using a monthly composite day approach, where all meteors detected during one month are treated as if they occurred on the same day. There may not be enough meteor echoes to reliably fit winds during local afternoon on any one day, but over a month there are. We found at least several hundreds of echoes within 1 standard deviation of every bin centre at 100km altitude during a month of afternoons throughout the year, which is more than enough to reliably fit a wind direction to. The rightmost panels of Fig. 4 show the same approach but include meteors during each month from all years 2016-2020, which increases the number of meteors available by a factor of 4-5.

Further still, instead of taking the simple average of the 24 wind measurements for each height during one composite day, we use the residual least-squares fitted mean from the sinusoidal tidal fit (see the background wind term $C$ in the tidal fit described in response to reviewer #1 above and now included in the revised manuscript). This is the least-squares fitted mean value around which the fitted sinusoidal tidal perturbations oscillated, which is the background wind. This approach again can cope with one or two missing data points if the situation arises (which does not occur anyway for a composite day of a whole month) and means that we avoid biases from averaging over incomplete data.

We also considered the possibility that meteors close to 90km altitude were driving the fit at higher altitudes, so we repeated the analysis for Fig. 4 but removed any meteors below 100km altitude. We got the same result: equatorward winds above 100km during summer.

We apologise for the exhaustive response, but we wanted to show that we have considered all angles because it is one of our more significant conclusions.

4. **Sect. 3.2** *Is there a regular bias between the two methods at the heights where both of them provide data? Maybe you can add a figure in the supplement.*

   Thanks for this suggestion. Winds derived using the height gates method are, for the most part, simply a coarser version of the Gaussian-weighting derived winds and we did not find any specific biases in derived winds. We do find however that the traditional binning method is more prone to spurious values for regions with low meteor counts than the Gaussian weighted version, because in the latter neighbouring meteor echoes can be used to constrain the wind fit which is unlikely to vary wildly from one hour to the next.

   We found that the most significant difference between the two methods is the recorded time and height to which the derived winds are assigned. The difference in altitude can be up to around 1 km in height and up to 20 minutes in time. Correct measurement times for derived winds are very important if we are trying to accurately fit tides, especially for higher order tidal modes. This is another advantage of the more accurate time values provided by the new method.

   As the reviewer suggested, we have included a figure showing the time and height differences between the "bin centres" and the Gaussian-weight times and heights of the meteors used to derived the wind in two extra panels on Fig. 2. We thought that this was quite an important point, so we are happy to include it in the main body of the revised manuscript.

5. **Fig. 3** *The S4 peak is rather well visible. I would be interested if there is a seasonal cycle in 6hr tidal activity, even if the amplitudes amount to just a few m/s.*

   We are very grateful for this comment - we have now included and discussed the S4 tide in our presented results in the revised manuscript. As the reviewer suspects, its amplitudes are only a few m/s, but there is some reasonable seasonal cycle in phase which suggests some confidence in the analysis. We are happy to include it.

6. **Fig. 3, l.202** *could you add a -5/3 line to the figure? It looks like there is a slight change of the slope at about 4 hrs.*

   We have added a -5/3 line on to both panels of Fig. 3. We agree with the reviewer and we suspect that this slight change in slope is due to the GW signal falling detectable noise levels below periods of 4 hours. Although our winds are sampled hourly (which implies a 2 hour Nyquist resolution limit), because we use a sliding Gaussian window with a FWHM equal to 2 hours we are unlikely to resolve periods near 2-4 hours, so any oscillations at periods shorter than this are likely over contaminated by noise and measurement error. We have included this information in the manuscript.

7. **l.214** $54°$ *is frequently called "higher midlatitudes" rather than high latitudes, e.g. in Stober et al. 2021b.*

   Fixed, thanks.

8. **Fig. 4** *The summer meridional wind maximum is lower than the one seen at Tierra del Fuego in Stober et al. 2021b. Is there an explanation?*

   We assume the reviewer means lower in altitude, and yes we noticed this too. We wonder if there may be some significant quasi-stationary planetary wave activity (not detectable in our PW analysis) that could cause a difference between the apparent background meridional winds at the two sites. Without more information and a more in-depth study, we don't feel ready to comment further but have mentioned this difference in the revised manuscript.

9. **l.265** *terdiurnal -> terdiurnal*

   Fixed, thanks.

10. **l.335** *10 ms -> 10 ms-1*

    Fixed, thanks.

11. **l.427** *delete "that" after "limitations"*

    Fixed, thanks.

12. **l.945** *reference should read "Solomon, S., and R.R. Garcia"*

    Good catch, thank you! Fixed.

**Response to Reviewer #3**

**General Comments**

- *This manuscript presents comprehensive analysis of horizontal winds observed using the meteor radar deployed at South Georgia island over the Southern Ocean and temperature obtained from the MLS sensor mounted on the AURA satellite in terms of mean flow, tides, planetary waves and gravity waves. In addition, this study compares the analysis results with the climatological WACCM results and suggest directions in which whole atmosphere climate models like WACCM can be improved in the future for better MLT simulations.*

  Thank you, this is a good summary of the study.

- *The conventional methodology in retrieving horizontal winds from meteor radar echoes is overcome through the development of a novel technique based on Gaussian weighting approach. This new technique allow for obtaining the smoother wind information as much as possible from meteor radar observations. This study is impressive in that it includes almost every analysis that can be done with meteor radar horizontal winds. Most of descriptions of results are reasonable, but there is an issue that need to be clarified. This issue is related to conclusion of this study and may affect considerably discussions about secondary gravity waves. For this reason, reviewer recommends major revision for consideration of publication to Atmospheric Chemistry and Physics.*

  We are grateful that the reviewer appreciated our effort to provide a complete picture of the major dynamics of the MLT above South Georgia in this paper, including winds, tides, PWs and GWs.

  In the revised manuscript, we have removed the analysis and discussion of the directional GW tendencies that the reviewer discusses in their major and minor comments below. We removed this section mainly because we now believe that more investigation is needed to ensure that the results of the method are not influenced by errors in meteor height determination, although the reviewer's comments are what led us to consider this factor. For completeness, we respond to each of the reviewer's points below in any case, although all discussion of GW directionality and 2GW inference in our measurements has been removed.

  The most significant issue that we discovered with the directional tendency method relates to error in the height finding of individual meteor echoes. At high zenith angles, a relatively small error in a meteor echo's measured zenith angle can have a relatively large error in the altitude that we assign to that meteor echo. This is especially a problem during summer, when there is a strong zonal wind shear of around 4-5 $ms^{-1}$ per km over the meteor radar's detectable height range. A small error of a few km in the height assigned to a meteor echo in the zonal direction during summer could result in a wind measurement that is quite different from the mean horizontal wind at that level, leading to a large wind variance measurement. This effect is especially significant for the zonal direction during summer, since the

zonal wind shear with height during winter and meridional wind shear all year is much less significant (see our Fig. 4). This could have an impact of the increased wind variance in the zonal direction during summer, as originally reported in our Fig. 7e-f. For this reason, we decided that although our results were consistent with those of Manson et al. (2004), further testing was needed to separate this issue from any geophysical differences in directional variances due to GWs.

The resolved and sub-volume GW variance methods are not as sensitive to this issue because they consider the whole radar field of view, so they have remained in the revised manuscript and are consistent with previous studies.

We are grateful to the reviewer for suggesting that we take a closer look at the directional tendency method and consider alternative sources of error.

**Major Comments**

1. *One major issue is about inferring the propagation direction of gravity waves derived from the sub-volume gravity-wave variances. Authors said in the paragraph between L490 and L496 that any residual meteor wind perturbation due to this (zonally propagating) gravity wave measured to the east or west of the radar will exhibit larger radial wind perturbations than those measured to the north or south. Therefore, if the variance of the residual perturbations in the zonal direction is larger than the variance in the meridional direction, we can infer that there is a zonal "directional tendency" of GWs towards the zonal direction.*

   The reviewer has correctly summarised the fundamental assumptions of the method, although as mentioned above we have removed these results from the revised manuscript.

   For completeness in our responses below we provide more explanation of the method in case we consider it for future use, although for the reasons we listed above we no longer include it in the revised manuscript.

   - *First of all, could you confirm the formulations of $\sigma_{zon}^2$ and $\sigma_{mer}^2$? Are they defined by something like the following equations? $\sigma_{zon}^2 = \left(\Sigma_{i=1}^N v_{ri}'^2 \sin^2 \theta_i\right)/(N-1)$ and $\sigma_{mer}^2 = \left(\Sigma_{i=1}^N v_{ri}'^2 \cos^2 \theta_i\right)/(N-1)$, where $v_{ri}'$ is the residual radial (horizontal) wind at each radar echo position, $\theta_i$ is the azimuth angle of the radial (horizontal) wind measured clockwise from the North. Here, the residual wind is the result of subtracting radial projection of the hourly zonal and meridional winds from the observed radial winds on the horizontal plane as in Mitchell and Beldon (2009).*

     Yes that is the general method, but with a few key differences. $v_{ri}'$ is indeed the residual radial velocity at each radar echo position $i$, once the radial projection of the large-scale horizontal winds has been subtracted as in Mitchell and Beldon (2009). Assuming zero mean (correct for perturbations from the background wind), the weighted variance $\sigma_{\mathrm{zon}}^2$ is given by

     $$\sigma_{\mathrm{zon}}^2 = \frac{\sum_{i=1}^N w_i\, v_{ri}'^2}{\frac{(N-1)}{N} \sum_{i=1}^N w_i} \tag{2}$$

     where $N$ is the number of meteor echoes and $w_i$ is a weighting for each echo given by $w_i = \sin^2 \theta_i$. For $\sigma_{\mathrm{mer}}^2$, the formula is the same but the weighting is $w_i = \cos^2 \theta_i$. The reviewer will notice that we did originally use the modulus of $|\sin \theta_i|$ and $|\cos \theta_i|$ as our weighting functions in the original manuscript, but we have revised the method now to use the square of the functions as they suggested. Using the squared functions results in a better localisation of perturbations in the zonal and meridional directions, so we thank the reviewer for their comment.

     As mentioned to reviewer #1, this weighted variance approach is very similar to simply binning the measured echoes into azimuth bins aligned north-south and east-west and calculating the variance in each bin. The weighting approach is however a little more elegant and allows us to use all available meteor echoes with no hard cut-offs at the bin edges.

- *If the two formulations are what the authors employed, the directionality obtained based on these equations may give biased estimate of the propagation direction of gravity waves because $\sin^2$ $(\cos^2)$ have the same positive weights for eastward and westward (northward and southward) propagating gravity waves. Figures below show possible situations of the phase progressions for gravity waves around a radar site. On the first panel, upward propagating gravity waves propagate eastward (or northward) relative to the zonal (meridional) mean flow. Radar observation is sensitive to those gravity waves east (north) of the radar, but not sensitive to those waves west (south) of the radar. On the second panel, upward propagating gravity waves propagate westward (southward) relative to the mean flow. Radar observation is sensitive to those gravity waves west (south) of the radar, but not sensitive to waves east (north) of the radar. In other words, radar observations can be sensitive to gravity waves that are generated from the radar site and propagate in radial directions away from the radar. However, for gravity waves passing through the radar site, radar can detect those waves only either east or west (either north or south) of the radar site. In summary, the directionality computed in the manuscript can possibly be biased towards gravity waves generated near the radar site (or downward propagating toward the radar site in case of upward phase propagation).*

We thank the reviewer for their comment which made us take a closer look at potential sources of error in the directional tendency method.

The reviewer is suggesting here that the directional variance method as described may have different sensitivities towards the same GW on opposite sides of the radar, where wind perturbations due to the GW will disappear from view when the phase fronts are aligned parallel to the line of sight. We do not contest that this effect can occur, but we believe it will occur so rarely in reality that it cannot significantly affect our results.

In the figure provided by the reviewer, GW phase fronts from the same wave packet are aligned either perpendicular or parallel to the radar's field of view. In the latter case, the angle of the GW phase fronts with respect to the vertical equals the zenith angles of the detected meteor echoes.

As can be seen in our Fig. 1, the majority of meteor echo detections occur between zenith angles between around 45 to 60 degrees. For the GW phase fronts in the reviewer's figure to be parallel to this, this implies a ratio of horizontal to vertical wavenumbers of between around $\frac{m}{k} \approx 1$ and $\frac{m}{k} \approx 2$. GWs with ratios of $m/k$ this small have periods around 10 minutes, which is only slightly larger that the lower limit for GW periods of around 5 minutes (Brunt-Väisälä frequency). Even using approximate definitions given by Fritts and Alexander (2003), these would be regarded as extremely high frequency GWs. To our knowledge, such short period GWs are not commonly observed in the MLT by radars and would be quite extraordinary. We would normally expect GWs visible to our radars to have $\frac{m}{k}$ ratios of at least 5-10 or greater, which gives periods of at least 30 minutes to 1 hour. Therefore, for the vast majority of the expected GW periods visible to the radar, this effect will be small and no GWs are expected to significantly disappear from view on one side of the radar but not the other.

We propose that what is much more important for the direction variance therefore is the *horizontal* orientation of the GW phase fronts, which will disappear from view with changes in meteor echo azimuth with respect to the GW phase fronts. Because the horizontal wind perturbations due to GWs are expected to be larger than their vertical perturbations, this azimuthal effect will have a much more significant impact on the measured GW variance than the vertical orientation.

We hope that these explanations of our reasoning are clear, but we want to especially thank the reviewer for taking the time to challenge the method because it highlighted other factors which we had not considered, as discussed above.

- *Reviewer wonders if the authors have considered the above-mentioned issue. If the directionality computed is biased towards gravity waves propagating horizontally outward from (or towards) the radar site, the directionality in the present manuscript should be interpreted differently. Correction of the interpretation of the directionality may lead to modification of interpretation of the seasonal variations of the propagation direction of gravity waves*

*and modification of discussion of generation of secondary gravity waves.*

See our responses above. We have removed the analysis and any discussions of suggested 2GW directionality arising from the method.

**Specific Comments**

1. **l.11** *Symbols W1 and W2 are used without appropriate definitions.*

   Fixed, thanks.

2. **l.41** *Solomon and Garcia, 1987*

   Fixed, thanks.

3. **l.52** *Considering oblique propagation of GWs in parameterizations has a relatively long history, although Kalisch et al. (2014) is relatively recent one. Consider citing some of relevant previous studies: Song and Chun (2008), Hasha et al. (2008), or Choi et al. (2009).*

   Agreed, happy to add the references as suggested.

4. **l.53** *Current GW parameterizations do neither include secondary GWs nor GWs that originate from the middle atmosphere (upper stratosphere or lower mesosphere) (e.g., Fairlie et al., 1990; Sato and Yoshiki, 2008).*

   Agreed, references added as suggested.

5. **l.220** *How large-amplitude Q6DW can be related to small impacts due to southern SSW in September 2019?*

   Agreed, this point was confusing in the text so we have removed it. We think the primary reason we don't present any obvious SSW wind changes in our results during 2019 is simply because we are looking at monthly timescales in Fig. 4. The study of Liu et al. (2021) explored the Q6DW activity during this period, but we do not go any further to suggest links Q6DW-SSW in this study.

6. **l.222** *It does not seem reasonable to discuss the Brewer-Dobson circulation (BDC) at the mesopause heights, although the BDC is expected to manage to reach somehow middle mesosphere as inferred from the stream lines shown in Seviour et al. (2012). Besides, results in Murgatroyd and Singleton (1961) were mostly shown below z = 80 km.*

   Agreed, we should consider the summer-to-winter pole flow in the MLT separately from the (mostly) tropical and midlatitude BDC in the troposphere and lower stratosphere. We have revised the text to reflect this.

7. **Fig. 5** *How about include labels for zonal and meridional winds in panels (or as panel titles)*

   No problem, labels added, thanks.

8. **l.301** *How can we explain the upward propagation of diurnal tides in the high-latitude regions where the Coriolis frequency is larger than diurnal frequency?*

   Numerical simulations and observational measurements of an upwardly propagating diurnal tide at midlatitudes is well established in the literature (e.g. Smith, 2012, and citations therein), but we agree that the upward propagation of the diurnal tides is likely to maximise in the tropics and would only be weakly detectable at high latitudes.

   In our Fig. 5 however, we find that the there is a negative change in the phase of the diurnal tide with increasing height during summer when tidal amplitudes are consistently larger than $5\,\mathrm{ms^{-1}}$ (late summer). This is indicative of an upwardly propagating tide, so it does appear than when amplitudes are large there is some apparent upward propagation that can be measured, but when amplitudes are low this consistent phase change breaks down. We have included this information in the revised manuscript. We have also included white hatching on areas in Fig. 5 where

the tidal amplitudes are small to make it clearer to the reader which regions of measured phases are less likely to be realistic due to small tidal amplitudes.

9. **l.399** *The subsection number should begin from 1.*

   Fixed, thanks.

10. **l.465** *100s -> hundreds*

    Fixed, thanks.

11. **l.532-537** *Secondary gravity waves with concentric-ring shaped phase lines can have meridional components of horizontal propagation even though they are generated by a zonal body force. That being said, logic used to explain the meridional GW propagation tendency seems weak because the major propagation direction of secondary GWs still lies in the sense of the direction of the body force induced by the primary GWs. This part is also related to the major issue raised above.*

    Agreed, see responses above. The directionality argument that inferred 2GW detection has been removed.

12. **l.535** *stratopause -> mesopause*

    Fixed, thanks.

13. **l.594** *WACCM observations -> WACCM*

    Fixed, thanks.

14. **Sect. 9** *Although authors suggest impacts of secondary GWs as a promising mechanism to improve the MLT simulations of whole atmosphere models based on recent studies (e.g., Becker and Vadas, 2018), there are also substantial uncertainties regarding impacts of GW generation due to flow imbalance in the upper stratosphere and lower mesosphere. Recommend to mention even briefly potential middle-atmospheric GW sources as mentioned above.*

    Thanks, this has been included in the text.

**References**

R. N. Davis, J. Du, A. K. Smith, W. E. Ward, and N. J. Mitchell. The diurnal and semidiurnal tides over ascension island (°s, 14° w) and their interaction with the stratospheric quasi-biennial oscillation: studies with meteor radar, eCMAM and WACCM. *Atmospheric Chemistry and Physics*, 13(18):9543–9564, September 2013. doi: 10.5194/acp-13-9543-2013.

D. C. Fritts and M. J. Alexander. Gravity wave dynamics and effects in the middle atmosphere. *Reviews of Geophysics*, 41: 1003, 2003. doi: 10.1029/2001RG000106.

Guiping Liu, Diego Janches, Ruth S. Lieberman, Tracy Moffat-Griffin, Nicholas J. Mitchell, Jeong-Han Kim, and Changsup Lee. Wind variations in the mesosphere and lower thermosphere near 60s latitude during the 2019 antarctic sudden stratospheric warming. *Journal of Geophysical Research: Space Physics*, 126(5), May 2021. doi: 10.1029/2020ja028909.

H.-L. Liu. On the large wind shear and fast meridional transport above the mesopause. *Geophysical Research Letters*, 34 (8), April 2007. doi: 10.1029/2006gl028789.

A. H. Manson, C. E. Meek, C. M. Hall, S. Nozawa, N. J. Mitchell, D. Pancheva, W. Singer, and P. Hoffmann. Mesopause dynamics from the scandinavian triangle of radars within the PSMOS-DATAR project. *Annales Geophysicae*, 22(2): 367–386, January 2004. doi: 10.5194/angeo-22-367-2004.

N. J. Mitchell and C. L. Beldon. Gravity waves in the mesopause region observed by meteor radar::1. A simple measurement technique. *JASTP*, 71(8-9):866–874, JUN 2009. ISSN 1364-6826. doi: {10.1016/j.jastp.2009.03.011}.

Liying Qian, Alan Burns, and Jia Yue. Evidence of the lower thermospheric winter-to-summer circulation from saber co2 observations. *Geophysical Research Letters*, 44(20):10,100–10,107, 2017. doi: https://doi.org/10.1002/2017GL075643.

Anne K. Smith. Global dynamics of the MLT. *Surveys in Geophysics*, 33(6):1177–1230, June 2012. doi: 10.1007/s10712-012-9196-9.

G. Stober, A. Kozlovsky, A. Liu, Z. Qiao, M. Tsutsumi, C. Hall, S. Nozawa, M. Lester, E. Belova, J. Kero, P. J. Espy, R. E. Hibbins, and N. Mitchell. Atmospheric tomography using the nordic meteor radar cluster and chilean observation network de meteor radars: network details and 3d-var retrieval. *Atmospheric Measurement Techniques*, 14(10):6509–6532, 2021a. doi: 10.5194/amt-14-6509-2021.

G. Stober, A. Kuchar, D. Pokhotelov, H. Liu, H.-L. Liu, H. Schmidt, C. Jacobi, K. Baumgarten, P. Brown, D. Janches, D. Murphy, A. Kozlovsky, M. Lester, E. Belova, J. Kero, and N. Mitchell. Interhemispheric differences of mesosphere–lower thermosphere winds and tides investigated from three whole-atmosphere models and meteor radar observations. *Atmospheric Chemistry and Physics*, 21(18):13855–13902, 2021b. doi: 10.5194/acp-21-13855-2021.

---

## Referee Report (RR1)

**Review of "Radar observations of winds, waves and tides in the mesosphere and lower thermosphere over South Georgia island (54°S, 36°W) and comparison to WACCM simulations"**

**General comments**

- Reviewer recommends this manuscript for publication to ACP. The authors' reply is convincing especially regarding the estimation of the horizontal propagation of gravity waves. It seems that the azimuthal distribution of horizontal radial velocity can be important in estimating the horizontal propagation direction of gravity waves observed within the field of view of meteor radars. Also reviewer appreciate authors' efforts to report mesospheric systematic biases that even the state-of-art high-top models like WACCM (CESM2) have. This report will help the community focus more on those issues. Thank you authors and the editor for having chance to read this good paper on meteor radars and gravity waves.

---

## Author Response (AR2)

**Authors' response to final revisions on acp-2021-981: "Radar observations of winds, waves and tides in the mesosphere and lower thermosphere over South Georgia Island (54°S, 36°W) and comparison to WACCM simulations."**

N. P. Hindley et al.

We would once again like to thank the reviewers for their time and careful consideration of our manuscript, their helpful comments and suggestions have greatly improved this work.

**Response to Reviewer #1**

Accept subject to technical corrections:

- *Re: Sub volume GWs (line 468-479): It is worthwhile to comment why N and E "sub-variances" weren't done (presumably one reason is the shortage of meteors in the near-North and near-East directions.)*

  We have added a short note in the discussion section describing why the directional sub-variances were not included in the final manuscript. As discussed in our first response, low daily meteor counts are indeed a problem but this was not the main problem for our approach (we only considered rolling monthly time windows). The main reason we removed the analysis was because of the strong summertime zonal wind shear in the MLT which, when combined with error in the meteor height determination, can lead to increased error in the subtraction of the local background wind from the radial velocity of each meteor. This leads to an artificial increase in the apparent variance of these perturbations in the zonal direction during summer that we might wrongly attribute to GWs orientated in the zonal direction. Or rather, we cannot currently separate the two causes of increased variance with confidence.

  In theory, the method should work well throughout the rest of the year, but we have included the note in the discussion describing this issue for residual perturbations in the zonal direction during summer.

- *Figure 7: The headers for the top two panels are obscured by the heavy red. The bottom (ERA5) headers are almost unreadable in light blue; as are the blue graphs.*

  Fixed, thanks. The headers are now white with a black outline, and the light blue has been changed to orange.

- *Line 607: "... that strong modulate ..."?*

  Fixed, thanks.

- *Line 654-5 looks like a "reminder" note to the author ?*

  Fixed. Thank you for catching this, this was indeed a direct reminder of one of the reviewers suggestions!

**Response to Reviewer #2**

No suggested revisions, accept as is.

**Response to Reviewer #3**

No suggested revisions, accept as is.